# Mapping 532 nm Lidar Ratios for CALIPSO-Classified Marine Aerosols using MODIS AOD Constrained Retrievals and GOCART Model Simulations

Travis D. Toth<sup>1</sup>, Marian B. Clayton<sup>2</sup>, Zhujun Li<sup>3</sup>, David Painemal<sup>1</sup>, Sharon D. Rodier<sup>4</sup>, Jayanta Kar<sup>3</sup>, Tyler J. Thorsen<sup>1</sup>, Richard A. Ferrare<sup>1</sup>, Mark A. Vaughan<sup>1</sup>, Jason L. Tackett<sup>1</sup>, Huisheng Bian<sup>5</sup>, Mian Chin<sup>6,7</sup>, Anne E. Garnier<sup>3</sup>, Ellsworth J. Welton<sup>6</sup>, Robert A. Ryan<sup>2</sup>, Gregory L. Schuster<sup>1,7</sup>, Charles R. Trepte<sup>1</sup>, and David M. Winker<sup>1</sup>

<sup>1</sup>NASA Langley Research Center, Hampton, VA

<sup>2</sup>RSES: Coherent Application, Inc. – Psionic LLC, Hampton, VA

<sup>3</sup>RSES: Analytical Mechanics Associates, Inc., Hampton, VA

<sup>4</sup>RSES: ADNET Systems, Inc., Hampton, VA

<sup>5</sup>University of Maryland Baltimore County, Baltimore, MD

<sup>6</sup>NASA Goddard Space Flight Center, Greenbelt, MD

<sup>7</sup>Retired

Correspondence to: Travis D. Toth (travis.d.toth@nasa.gov)

Abstract. The NASA-CNES Cloud-Aerosol Lidar and Infrared Pathfinder Satellite Observations (CALIPSO) mission provided a spaceborne global record of atmospheric aerosol and cloud profiles from June 2006 to June 2023. As an elastic backscatter lidar, the CALIPSO Cloud-Aerosol Lidar with Orthogonal Polarization (CALIOP) typically required an assumption of the aerosol lidar ratio (extinction-to-backscatter ratio; Sa) to retrieve aerosol extinction and column-25 integrated aerosol optical depth (AOD). In all previous versions of its data products, the CALIPSO extinction algorithms first determine the aerosol types then assign one Sa value globally for each aerosol type (e.g., 23 sr for marine at 532 nm). One of the major changes for the final CALIPSO data products release (Version 5, or V5) is the implementation of regional and seasonal Sa tables for CALIOP-classified "marine" aerosols. In this study, we describe the process of creating the 30 tables using 12 years (June 2006-August 2018) of Aqua Moderate Resolution Imaging Spectroradiometer (MODIS) total column AODs to constrain collocated CALIOP backscatter profiles in a Fernald inversion scheme and infer Sa (at 532 nm), focusing solely on the CALIOP "marine" aerosol type. The Goddard Chemistry Aerosol Radiation and Transport (GOCART) global aerosol model is used to estimate sea salt volume fraction (SSVF) that are collocated with the constrained 35 S<sub>a</sub> retrievals. Patterns of smaller SSVF (< 65%) and larger constrained Sa (> 40 sr) are found near land masses, while larger SSVF (> 95%) and smaller constrained S<sub>a</sub> (< 30 sr) are generally observed in the remote oceans. The inverse empirical relationship found between modeled SSVF and constrained Sa over global oceans yields values of ~21 sr for SSVF of 100% (i.e., "pure" marine) and ~58 sr for SSVF of 0% (i.e., the absence of marine aerosol). This relationship is 40 applied to develop regional and seasonal hybrid (retrieval and model-assisted) climatological Sa maps for CALIOP-classified marine aerosols; i.e., when MODIS-constrained results are not available, the model-assisted values are used. These hybrid Sa maps are subsequently used to retrieve new CALIPSO Level 2 (L2) aerosol extinction profiles and column AODs in the V5 release. For a 4-month (January, April, July, and October 2015) analysis, the V5 L2 CALIPSO AODs compared better to CALIPSO Ocean Derived Column Optical Depth (ODCOD) than the 45 CALIPSO Version 4.51 (V4.51) standard AODs in several regions, most notably the Bay of Bengal/Arabian Sea, where smoke/pollution typically mixes with marine aerosols. Also, the V5 CALIPSO AODs likely provide a lower AOD bias and root-mean-square-error than V4.51 AODs relative to coastal and island Aerosol Robotic Network (AERONET) AODs, as found in a validation study using data from June 2006 through October 2022. The technique described in this study contributes to CALIPSO's final V5 data products release and provides critical S<sub>a</sub> information for future spaceborne elastic backscatter lidars.

#### 1. Introduction








Acquiring observations since June 2006, the Cloud-Aerosol Lidar with Orthogonal Polarization (CALIOP) instrument aboard the NASA-CNES Cloud-Aerosol Lidar and Infrared Pathfinder Satellite Observations (CALIPSO) satellite delivered a long-term (~17 year) global record of vertical profiles of Earth's atmosphere (Winker et al., 2010) before ceasing operations in June 2023. CALIOP measured the vertical structure of atmospheric aerosols and clouds, providing critical information about their many roles in the Earth's radiation budget (e.g., Kato et al., 2011; Thorsen et al., 2017) and air quality (e.g., Kar et al., 2015; Toth et al., 2014; 2019; 2022). As an elastic backscatter lidar system, CALIOP directly measured range-resolved profiles of attenuated backscatter coefficients at 532 nm and 1064 nm. To retrieve extinction coefficients, unattenuated backscatter, and optical depths (i.e., height integration of extinction coefficient), which are the primary quantities of interest for a variety of applications in the scientific community, elastic backscatter lidars generally need additional information and/or assumptions regarding the lidar ratio (S<sub>a</sub>) - i.e., the ratio between particulate extinction and backscatter coefficients – and assume that the Sa remains constant throughout the vertical extent of any layer (e.g., Spinhirne et al., 1980; Ackermann et al., 1998). The S<sub>a</sub> is an intensive parameter that depends on several microphysical factors, including composition, size, shape, and refractive index (e.g., Ackermann et al., 1998), and thus varies according to aerosol type or species (e.g., Burton et al., 2012; Floutsi et al., 2023).

The  $S_a$  values used in the CALIOP aerosol retrieval algorithms are based on the tropospheric aerosol types derived via a cluster analysis using Aerosol Robotic Network (AERONET) data (Omar et al. 2005), from which CALIPSO's six original aerosol types were defined. At each wavelength, each aerosol type is assumed to be characterized by a single, globally constant  $S_a$  paired with a fixed standard deviation that describes the  $S_a$  natural variability within the type (Omar et al., 2009). For the "clean marine" type, a value of 20 sr  $\pm$  6 sr at 532 nm was

chosen based on measured size distributions of hydrated marine aerosols acquired during the Shoreline Environment Aerosol Study (SEAS) (Masonis et al., 2003). The value of 20 sr for clean marine was retained through CALIPSO Version 3 (V3) but was updated to a value of 23 sr in Version 4.10 (V4.10), such that CALIPSO's standard marine S<sub>a</sub> was more consistent with measurements made during a number of field campaigns(Kim et al., 2018). These include the Second Aerosol Characterization Experiment (ACE 2; e.g., Ansmann, 2001), Indian Ocean Experiment (INDOEX; e.g., Welton et al., 2002), and airborne High Spectral Resolution Lidar (HSRL) underflights of CALIPSO (e.g., Rogers et al., 2014). Relevant details of these campaigns are found in Table 2. Note that several studies reported lower marine aerosol optical depths (AODs) for CALIPSO compared to Moderate Resolution Imaging Spectroradiometer (MODIS; e.g., Oo and Holz, 2011), Synergized Optical Depth of Aerosols (SODA; e.g., Dawson et al., 2015), and HSRL (e.g., Rogers et al., 2014). These discrepancies were at least partly attributed to the assignment of incorrect S<sub>a</sub>, including through possible aerosol misclassification.

Also, while there were only six CALIPSO aerosol types through V3, the V4.10 release introduced a seventh aerosol type: dusty marine. This type was added to account for mixtures of marine and dust aerosol occurring over the oceans, especially Saharan dust during transport across the Atlantic Ocean (e.g., Liu et al., 2008; Groß et al., 2016; Kuciauskas et al., 2018). In V3, these features would typically be classified (incorrectly) as polluted dust, as airborne HSRL measurements of  $S_a$  for CALIPSO "polluted dust" aerosol layers (~35 sr) suggest a mixture of dust and marine as opposed to that of dust and smoke (Burton et al., 2013). Kim et al. (2018) report that the frequency of the polluted dust aerosol type over oceans significantly decreases with the introduction of the new dusty marine type. The characteristic  $S_a$  for dusty marine,  $37 \pm 15$  sr, was computed from the dust (44 sr) and clean marine (23 sr)  $S_a$  by assuming a dust to clean marine mixing ratio of 65:35 (by surface area). Table 1 shows the V4.10 CALIPSO  $S_a$  values, and estimated uncertainty ranges, for each of the seven CALIPSO tropospheric aerosol types. The  $S_a$  at 532 nm range from 23 sr (marine) to 70 sr (polluted continental/smoke and elevated smoke). These same values continued to be used through the release of CALIPSO's Version 4.51 (V4.51) data products.

**Table 1.** S<sub>a</sub> and corresponding estimated uncertainties (in units of sr) at 532 nm for each tropospheric aerosol type in the CALIPSO Version 4 algorithms (adapted from Kim et al. 2018).

| Tropospheric Aerosol Type  | V4 532 nm<br>S <sub>a</sub> (sr) |
|----------------------------|----------------------------------|
| Marine                     | 23 ± 5                           |
| Dusty Marine               | $37 \pm 15$                      |
| Dust                       | 44 ± 9                           |
| Polluted Dust              | $55 \pm 22$                      |
| Clean Continental          | 53 ± 24                          |
| Polluted Continental/Smoke | $70 \pm 25$                      |
| Elevated Smoke             | $70 \pm 16$                      |

The V4.51 tropospheric aerosol classification algorithm (Fig. 1) uses a number of parameters, including CALIOP estimated particulate depolarization ratio (EPDR), surface type, CALIOP 532 nm integrated attenuated backscatter (IAB), and CALIOP layer height. The CALIOP marine aerosol classification requires an aerosol layer to be detected over water, with its top altitude  $\leq 2.5$  km, and either an IAB > 0.01 sr<sup>-1</sup> and EPDR < 0.075, or IAB  $\leq 0.01$  sr<sup>-1</sup> and EPDR 

**Figure 1.** Flowchart of the CALIPSO Version 4 tropospheric aerosol classification algorithm and  $S_a$  selection process (Kim et al. 2018).  $\gamma'$  indicates 532 nm integrated attenuation backscatter (IAB),  $\delta_p^{\text{est}}$  indicates the estimated particulate depolarization ratio (EPDR), and  $Z_{\text{top}}$  and  $Z_{\text{base}}$  are the layer top and base altitude, respectively.

Sea salt aerosol is the primary aerosol species over the oceans and is generated by sea spray/bubble bursting through wave breaking (e.g., O'Dowd and De Leeuw, 2007). Marine aerosol, of which sea salt is the dominant component, also consist of a host of other aerosol species generated from natural and anthropogenic sources (e.g., Lewis and Schwartz, 2004). Due to the extensive coverage of oceans over Earth's surface, marine aerosol is a major component of the atmospheric aerosol composition near the surface (e.g., Murphy et al., 2019). In general, the size distribution of marine aerosol is dominated by the coarse mode, with some fine mode (e.g., Porter and Clarke, 1997; Yu et al., 2019). However, this can vary by the surface wind speeds, as higher speeds can lead to a greater number of larger particles. The resultant Sa for this scenario may tend to be smaller, as larger particles tend to exhibit smaller Sa (e.g., Masonis et al., 2003; Dawson et al., 2015). In addition to winds, relative humidity (RH) also affects marine aerosol size through particle hygroscopic growth, as higher RHs lead to larger particles, thus impacting the Sa (e.g., Ackermann et al., 1998). Also, in terms of the impact of sea salt sphericity on Sa, Haarig et al. (2017) found similar Sa for non-spherical and spherical sea salt aerosols using Raman lidar. A more recent study (Ferrare et al., 2023) arrived at a similar conclusion using HSRL measurements.

In this study, we investigate the regional and seasonal patterns of CALIOP-classified marine aerosol  $S_a$  with the goal of providing tables indexed by latitude, longitude, and season as an improvement over the single value currently used globally. We focus on aerosol classified as "marine" by CALIOP due to the large sample size of this aerosol type, and because a more robust MODIS AOD dataset exists over ocean compared to over land. For example, MODIS AOD retrievals over land are difficult due to the large variability in surface characteristics and exhibit larger uncertainties ( $\pm(0.05+15\%)$ ) than over ocean (( $\pm(0.04+10\%)$ ),  $\pm(0.02+10\%)$ ) (Levy et al., 2013). This over-ocean MODIS AOD dataset provides a critical component of this study in creating the  $\pm(0.05+15\%)$  the next section).

Table 2. Literature review of S<sub>a</sub> (mostly at or near 532 nm) in marine environments.

| Study                  | S <sub>a</sub> (sr) | Wavelength<br>(nm) | Method/Technique         | Location                               |
|------------------------|---------------------|--------------------|--------------------------|----------------------------------------|
| Ansmann et al. (2001)  | 20-25               | 532                | Raman                    | Portuguese coast                       |
| Bohlmann (2018)        | 23 ± 1              | 532                | Raman                    | Atlantic Ocean                         |
| Breon et al. (2013)    | 25                  | 670                | POLDER                   | Remote global oceans                   |
| Burton et al. (2012)   | 20 ± 5              | 532                | HSRL                     | Caribbean Sea                          |
| Cattrall et al. (2005) | 28 ± 5              | 550                | AERONET inversion        | Various island sites                   |
| Dawson et al. (2015)   | 26                  | 532                | SODA AOD<br>& CALIOP IAB | Global                                 |
| Doherty et al. (1999)  | 21.1 ± 3.7          | 532                | Backscatter nephelometer | Shore of northwest<br>Washington state |
| Franke et al. (2001)   | < 30                | 532                | Raman                    | Indian Ocean                           |
| Groß et al. (2011b)    | 17-19 ± 2           | 532                | Raman                    | Cape Verde                             |

| Li et al. (2022)               | 24-28                  | 532 | Constrained Fernald inversion (SODA/CALIOP)     | Global                                       |
|--------------------------------|------------------------|-----|-------------------------------------------------|----------------------------------------------|
| Masonis (2003)                 | 25 ± 3.5               | 532 | In situ                                         | East coast of Oahu,<br>Hawaii                |
| Müller et al. (2007)           | 23 ± 5                 | 532 | Raman                                           | North Atlantic and<br>Indian Oceans          |
| Papagiannopoulos et al. (2016) | $23 \pm 3$             | 532 | Raman                                           | Various European sites                       |
| Pedros et al. (2009)           | 31-37                  | 532 | Sun<br>photometer/aerosol<br>model inversion    | North Atlantic Ocean                         |
| Rittmeister (2017)             | 17 ± 5                 | 532 | Raman                                           | Atlantic Ocean                               |
| Rogers et al. (2014)           | 27 ± 14                | 532 | HSRL                                            | Caribbean Sea; mid-<br>Atlantic coast of US  |
| Sayer et al. (2012)            | 24 - 33                | 532 | AERONET inversion                               | Various island sites                         |
| Schmid (2003)                  | 34                     | 523 | Constrained Fernald inversion (MPL)             | Coast of Japan                               |
| Smirnov et al. (2003)          | 34.5                   | 500 | AERONET inversion                               | Lanai, Hawaii                                |
| Voss (2001)                    | $32 \pm 6$ $36 \pm 16$ | 523 | Constrained Fernald inversion (MPL)             | North Atlantic Ocean<br>South Atlantic Ocean |
| Wang (2020)                    | 30 ± 12                | 527 | Constrained Fernald inversion (MPL)             | Northern Taiwan                              |
| Welton (2002)                  | 33 ± 6                 | 523 | Constrained Fernald inversion (MPL)             | Indian Ocean                                 |
| Young et al. (1993)            | > 30                   | 532 | Backscatter lidar<br>(horizontally<br>oriented) | Coast of northern<br>Australia               |

A number of studies have investigated marine S<sub>a</sub> through a variety of instruments and methods, some global in scale and others focusing on specific oceanic regions (Table 2). One global analysis, Dawson et al. (2015), derived S<sub>a</sub> using SODA AOD and CALIOP IAB, and

segmented results as a function of surface wind speeds from the Advanced Microwave Scanning Radiometer – EOS (AMSR-E). A global mean S<sub>a</sub> over oceans of 26 sr was found, with a wind dependence on the S<sub>a</sub> values derived (e.g., ~32 sr for wind speeds less than 4 ms<sup>-1</sup> but ~22 sr for wind speeds greater than 15 ms<sup>-1</sup>). Another global study, Li et al. (2022), used SODA AOD to constrain the CALIOP backscatter profiles and derive S<sub>a</sub> using a Fernald inversion scheme (Fernald 1972; 1984) similar to the one used for this work. Li et al. (2022) further segmented these derived S<sub>a</sub> as a function of CALIOP aerosol type. They found global CALIOP-classified marine 532 nm S<sub>a</sub> values of 24-25 sr (medians) and 26-28 sr (means). A spatial pattern in S<sub>a</sub> was also found, with lower S<sub>a</sub> in the remote oceans, and higher values near coasts (e.g., Bay of Bengal and Arabian Sea). This was attributed to CALIOP misclassifying these features as marine rather than a mix of marine aerosol and pollution. A similar spatial pattern in S<sub>a</sub> is found in this study(Sect. 4).

Some studies have used shipborne Micropulse lidar (MPL) backscatter profiles (at 523 nm), constrained by AOD from Microtops handheld sunphotometers, to derive over-ocean  $S_a$  from an inversion technique (Voss et al., 2001; Welton et al., 2002, Schmid et al., 2003; Groß et al., 2011b; ). A more recent study, Wang et al. (2020), retrieved  $S_a$  from measurements acquired at "a rural site with no significant near-source emissions" in northern Taiwan using backscatter profiles (at 527 nm) from the Micropulse Lidar Network constrained by AERONET AOD.  $S_a$  values were 30 sr  $\pm$  12 sr when the aerosol source was marine (i.e., advection from the Pacific Ocean), but were notably higher (39 sr  $\pm$  16 sr) when the aerosol source is from the Asian continent (i.e., pollution).

Other studies have used Raman lidars (e.g., Franke et al., 2001; Müller et al., 2007; Ansmann et al., 2001; Rittmeister et al., 2017; Bohlmann et al., 2018; Groß et al. 2011b; Papagiannopoulos et al., 2016) and backscatter lidars (e.g., Young et al., 1993) to investigate S<sub>a</sub> in marine environments.

HSRLs can directly measure S<sub>a</sub> and thus have also been used to study marine SAs. Burton et al. (2012) found 532 nm S<sub>a</sub> in the 15-25 sr range over the Caribbean Sea from airborne HSRL measurements. Using coincident HSRL/CALIOP profiles acquired during CALIPSO calibration validation studies, Rogers et al. (2014) found that, for aerosol layers classified by CALIOP as 'marine', HSRL measured 532 nm S<sub>a</sub> of ~26 sr during daytime, ~28 sr at nighttime, and ~27 sr for

daytime and nighttime combined. Note that the histograms of Rogers et al. (2014) show a pronounced peak for marine  $S_a$  in the low 20s sr, with a small number of outliers that skew the average to larger values. This suggests that "clean marine" exhibits a fairly stable value but that the  $S_a$  of the marine boundary layer (MBL) can be raised if continental aerosol mixes into it.







There are also non-lidar techniques that can be used to derive S<sub>a</sub>. For one, inversions using column-integrated aerosol observations can be employed to retrieve S<sub>a</sub> estimates (e.g., Smirnov et al., 2003; Cattrall et al., 2005; Sayer et al., 2012; Pedros et al., 2009; Breon et al., 2013). Secondly, marine S<sub>a</sub> information has been estimated from *in situ* backscatter nephelometer measurements, like those observed at the Cheeka Peak Observatory in the northwest corner of Washington State (Doherty et al., 1999), and on the east coast of Oahu, Hawaii during the Shoreline Environment Aerosol Study (SEAS) campaign (Masonis et al., 2003).

These studies illustrate that S<sub>a</sub> measured over the ocean vary spatially and temporally, providing additional motivation for the creation of S<sub>a</sub> tables that vary by region and environmental conditions. The extensive data record of CALIOP allows us to also construct Sa tables that vary seasonally. The overall goal of this study is the creation of regional and seasonal climatological Sa maps for CALIOP-classified marine aerosol by leveraging MODIS AOD retrievals to derive Sa estimates from collocated CALIOP attenuated backscatter profiles. When the data yield is insufficient, we augment our maps using S<sub>a</sub> estimated from sea salt volume fraction (SSVF) computed using global aerosol model simulations from the Goddard Earth Observing System (GEOS) Goddard Chemistry Aerosol Radiation and Transport (GOCART). We develop a combined observational/model dataset from June 2006 (first CALIOP observations) to August 2018, when CALIPSO left the "A-Train" satellite constellation to join CloudSat in the "C-Train", thereby terminating continuous collocation with Aqua MODIS observations. The newly developed Sa tables (by region and season) are then used to retrieve CALIPSO V5.00 aerosol extinction profiles and tropospheric AODs. These are compared against AODs from an independent CALIOP retrieval algorithm, the Ocean Derived Column Optical Depth (ODCOD; Ryan et al., 2024), and against AODs from island/coastal AERONET sites (Holben et al., 1998) following Thorsen et al. (2025). The purpose of this paper is to document the approach used to develop the marine Sa tables and improve aerosol retrievals in the final CALIPSO data products release (V5). Sa tables have also been developed for the dusty marine CALIOP aerosol type using similar methods, but here we focus solely on marine. Note that the aerosol classification algorithm

differentiating marine and dusty marine is unchanged between the V4.51 and V5 CALIPSO datasets. Only the S<sub>a</sub> assignment for these aerosol types has been modified for V5.

The remainder of the paper is organized as follows. Sect. 2 discusses the various remote sensing datasets used. Sect. 3 discusses the methods employed for this study. Sect. 4 provides the results of the work, including analyses of the constrained S<sub>a</sub>, modeled SSVF, development of the seasonal S<sub>a</sub> climatologies, validation efforts of incorporating these S<sub>a</sub> in the retrieval of CALIOP tropospheric AODs, and a case study over the Bay of Bengal and Arabian Sea. A summary of the study, ongoing work, and implications for future spaceborne elastic backscatter lidars are discussed in Sect. 5.

#### 2. Datasets

## 2.1 CALIPSO CALIOP





230

225

We utilize CALIPSO Version 4.51 (V4.51) data, with data release dates beginning September 2022. Specifically, 532 nm total attenuated backscatter profiles were taken from the V4.51 Level 1 files (CAL LID L1-Standard-V4-51). The "Feature Classification Flags" that provide high level characterization of CALIOP's L2 layer detection and classification results were taken from the corresponding V4.51 L2 vertical feature mask (VFM) product (CAL LID L2 VFM-Standard-V4-51). The VFM product was used for identifying cloud free single shot profiles in each 5 km data segment and determining aerosol top heights during the constrained retrieval process. Further, the V4.51 5 km aerosol profile product (CAL LID L2 05kmAPro-Standard-V4-51), specifically the "Atmospheric Volume Description" parameter, was used for partitioning the datasets by aerosol subtype and spatial averaging (i.e., averaging required for feature detection). The L3 stratospheric aerosol product (CAL LID L3 Stratospheric APro-Standard-V1-00) was used to obtain the stratospheric AOD ("Stratospheric Optical Depth" parameter). These stratospheric AODs are reported monthly at 5° x 20° latitude/longitude resolution and were constructed using only high-quality CALIOP nighttime data (Kar et al., 2019).

## 2.2 Aqua MODIS

The MODIS instruments, flying aboard the Terra (since 1999) and Aqua (since 2002) satellites, are passive sensors that provide column AOD retrievals at various wavelengths (Remer et al., 2005). CALIPSO flew in the "A-Train" satellite constellation with Aqua from June 2006 until September 2018 (i.e., until CALIPSO exited to join CloudSat in the "C-Train" orbit), so for over a decade the two sensors flew within a few minutes of one another, providing numerous opportunities for retrieval synergies and multi-sensor data fusion (e.g., Burton et al., 2010; Braun et al., 2019; Fujishin et al., 2024). MYD03 Geolocation 1 km files from the Collection 6.1 (C61) MODIS data release (Levy et al., 2013; Sayer et al., 2014) were used for collocation with CALIOP in this study (Sec. 2.2). The "Effective Optical Depth Best Ocean" parameter, from the matching L2 MYD04 10 km C6.1 MODIS files, is provided at four wavelengths (470, 550, 660, and 860 nm) and these were interpolated to the CALIOP visible wavelength of 532 nm through an Ångström relationship (Schuster et al., 2006) to be used in the constrained retrieval process. MODIS AODs exhibit uncertainties over land of ±(0.05+15%) and over ocean of (+(0.04+10%), -(0.02+10%) (Levy et al., 2013).

## 3. Methods

## 3.1 Constrained Sa Retrieval Primer

The constrained S<sub>a</sub> retrieval method used in this paper is similar in principle to the procedure used in Li et al. (2022). CALIOP Level 1 (L1) attenuated backscatter profiles with a nominal horizontal resolution of 5 km were created by averaging all cloud-free single shot (333 m) profiles detected within 15 consecutive shots. The optical depths ascribed to these profiles are retrieved from collocated MODIS AOD data that are corrected for stratospheric contributions using the CALIOP Level 3 (L3) stratospheric aerosol product (Kar et al., 2019). S<sub>a</sub> are retrieved for each 5 km profile by the iterated application of a Fernald solution. Beginning with an initial guess, S<sub>a</sub> are repeatedly adjusted until the integrated Fernald solution yields an optical depth that is essentially identical to the external MODIS+CALIOP constraint. The CALIOP Level 2 (L2) products are then queried to identify those profiles in which only a single aerosol type has been detected, such that we can restrict our analysis to solely CALIOP-classified "marine" aerosols. Detailed mechanics of the retrieval scheme are given in Sect. 3.2.

#### 3.2 Methods in Detail



Figure 2. Schematic of the overall approach for this study (2006-2018). The CALIPSO Level 2 vertical feature mask is used to find 5 km columns containing only marine aerosols, with at least some of the aerosol being detected using only 5 km spatial averaging. We assume all tropospheric AOD occurs within 2 km of the aerosol layer top (Li et al., 2022) and that "clear air" (i.e., no aerosol) exists from this altitude upward to the stratosphere. We subtract the CALIPSO Level 3 stratospheric AOD (available at 5° x 20° latitude/longitude resolution, at monthly intervals, and nighttime only) from the Collection 6.1 Aqua MODIS total column AOD to constrain the CALIPSO Version 4.51 5 km Level 1 backscatter profiles in a Fernald inversion scheme (Fernald, 1972; 1984).

As the first step of this study, multiple years (2006-2018) of global daytime satellite measurements from the CALIPSO lidar L1 V4.51 (CAL\_LID\_L1-Standard-V4-51) and MODIS Aqua C6.1 MYD03 Geolocation 1 km and MYD04 10 km datasets were combined and individual measurements were collocated using the University of Wisconsin Space Science and Engineering Center collocations routine Collopak (Nagle and Holz, 2009). Next, we apply a constrained Fernald inversion to CALIOP attenuated backscatter profiles. In this procedure, an initial estimate of S<sub>a</sub> is adjusted by increasingly smaller increments until the change in S<sub>a</sub> from one iteration to the next is less than 0.0001 sr and the layer optical depth calculated using the refined value is within 0.0001 of the externally supplied optical depth constraint. The optical depth constraints in this

study are derived from collocated total column MODIS AOD corrected for stratospheric contributions using CALIOP L3 products. S<sub>a</sub> are allowed to vary over a range from –50 sr to 150 sr to capture a wide spectrum of S<sub>a</sub> and because the iterations for the Fernald retrieval were numerically stable for this range (determined through sensitivity studies). Note that this approach produces a negligible fraction of negative S<sub>a</sub> values (less than 0.05%), and our methodology minimizes the influence of these outliers by using median values when creating the S<sub>a</sub> maps (Sects. 3 and 4).







This passive AOD constrained lidar retrieval method has been successfully used in past studies (e.g., Ferrare et al., 2006; Burton et al., 2010; Kim et al., 2017; Kim et al., 2020). In this study, CALIOP L1 V4.51 backscatter profiles are cloud-cleared using information provided by the Feature Classification Flags from the CALIPSO VFM files then averaged to a 5 km horizontal resolution (i.e., 333 m backscatter profiles with clouds at any altitude are removed from the 15 shot average). The MODIS Effective Optical Depth Best Ocean parameter (at 470, 550, 660, and 860 nm), collocated with CALIOP as discussed previously, was interpolated to CALIOP's 532 nm wavelength using an Ångström relationship (Schuster et al., 2006). To ensure high quality Ångström interpolations we required positive values for all four MODIS AODs and rejected those cases flagged as "bad retrievals" by MODIS's Land Ocean Quality Flag. Since MODIS AOD represents aerosol loading for the entire atmospheric column and this study focuses on tropospheric aerosol Sa, the CALIPSO L3 Stratospheric Aerosol Profile Product (SAPP; Kar et al., 2019) was used to remove the contribution of stratospheric aerosols (i.e., stratospheric AOD) from the constraints used in the Fernald inversion scheme. The SAPP is produced on a monthly basis at a spatial resolution of 5° latitude × 20° longitude using only nighttime CALIOP measurements. Under the assumption that the distribution of stratospheric aerosol is diurnally invariant, a stratospheric AOD was assigned to each 5 km CALIOP profile through temporal and spatial collocation. This stratospheric AOD was then subtracted from the column MODIS AOD to obtain an AOD to use in the Fernald inversion. Also, it is assumed that all tropospheric AOD is found within 2 km above the highest detected aerosol top (determined by the CALIPSO VFM product), which results in the upper altitude limit during the Fernald retrievals of Sa (Fig. 2). This upper altitude limit was based on the SODA-CALIPSO work of Li et al. (2022), which determined the 2 km value through a past investigation of CALIPSO-SODA/airborne HSRL comparisons (Painemal et al., 2019) and further supported by a CALIPSO/airborne HSRL study (Burton et al., 2013).

Results of sensitivity studies of CALIPSO-SODA S<sub>a</sub> by varying this upper altitude limit are found in Li et al. (2022).

The Atmospheric Volume Description parameter in the aerosol profile data was used to obtain feature classification information, in addition to horizontal averaging required for feature detection (5 km, 20 km, or 80 km) and Feature Type QA (quality assurance) flags. The CALIOP profiles used in the Sa retrievals were restricted to those reporting only marine aerosols with the highest quality assurance classification (i.e., Feature Type QA=3). An additional filtering step involved including only those profiles in which at least part of the aerosol layer was detected at a 5 km horizontal averaging resolution. Levying this requirement yields four possible scenarios: marine aerosol detected only at 5 km, at 5 km and 20 km, at 5 km and 80 km, and at 5 km, 20 km, and 80 km. This "some 5 km" requirement was implemented based on discussions in Li et al. (2022) regarding the confidence of the CALIPSO aerosol classification as it relates to spatial averaging. Li et al. (2022) conclude that lower confidences should be assigned to longer averages (i.e., 80 km), because while the extended averaging is necessary to increase the signal-to-noise ratio (SNR) for the detection of tenuous aerosol layers, using these larger distances increases the likelihood of averaging over a heterogenous scene.

#### 4. Results







## 4.1 Developing the relationship between the MODIS AOD constrained $S_a$ retrievals and modeled sea salt volume fraction (1° x 1° latitude/longitude grid)

The goal of this study is to produce data driven and empirically derived S<sub>a</sub> maps over global oceans on seasonal scales. However, MODIS AODs are only available for daytime observations and have seasonally limited data coverage (due in part to glint regions with no MODIS AOD), which introduces large, periodic swaths of missing data in the retrieved S<sub>a</sub> maps. To mitigate this issue, we first leveraged the GEOS GOCART model to obtain a characterization of the amount of sea salt aerosol in a given region of the ocean and then used these estimations to examine their relationship with the available constrained S<sub>a</sub> retrievals. The GEOS GOCART model provides simulations of the dominant aerosol species found in the atmosphere, such as sulfate, carbon, dust, and sea salt (Ginoux et al., 2001; Chin et al., 2002, 2009, 2014; Colarco et al., 2010). The model accounts for aerosol emissions from anthropogenic and natural sources, surface wind speeds, advection, convection, and boundary layer turbulent mixing. The model is driven by the

meteorological reanalysis from the Modern Era Reanalysis for Research and Applications version 2 (MERRA-2) with the GEOS system, provided by the NASA Global Modeling and Assimilation Office (GMAO). In this study, we used the model version GEOS-i33p2 BASE simulations from 2006 to 2018 that are archived at the AeroCom server as part of the AeroCom Phase III model experiments (descriptions available at https://aerocom.met.no/experiments/UTLS/). These simulations are available at 1° x 1° horizontal grid spacing and 72 vertical layers with daily temporal resolution.







GOCART simulates aerosol properties and concentrations for various aerosol species, including the following with one dry size bin: sulfate (SO<sub>4</sub><sup>2-</sup>), ammonium (NH<sub>4</sub>+), black carbon (BC), brown carbon (BrC), and organic carbon (OC). Each of the carbonaceous aerosols include a hydrophobic and hydrophilic (aged) component. Other aerosol species are represented in the model by their size-aggregated bins, including nitrate (NO<sub>3</sub>; three size bins), dust (five size bins), and sea salt (five size bins). To obtain the specific volume (i.e., volume per unit mass) of each aerosol species at each vertical level, aerosol mass mixing ratios (in kg kg<sup>-1</sup>) were divided by their respective particle densities (in kg m<sup>-3</sup>), as provided in Collow et al., 2023. The specific volume fraction of sea salt aerosol within 2.5 km altitude from the surface was computed by summing the specific volume of sea salt aerosol (Z

**Figure 3.** Twelve-year (2006-2018) (a) spatial median of S<sub>a</sub> retrievals and (b) corresponding number of samples per grid box, at 1° x 1° latitude/longitude resolution during daytime for profiles with only CALIOP-classified marine aerosols. Medians and samples are shown only for those grid boxes with at least 9 points and S<sub>a</sub> relative standard error (RSE) less than or equal to 10%.





Figure 3a shows the global spatial distribution of median 532 nm constrained  $S_a$  for the entire twelve-year (2006-2018) dataset. The corresponding sampling map is shown in Fig. 3b. Each grid cell reports results obtained from daytime CALIOP profiles in which only marine aerosol was detected and further filtered for sampling ( $\geq 9$  points) and RSE ( $\leq 10\%$ ). Note the lack of  $S_a$  retrievals in the high latitudes north of 60° or south of -60°, which occur due to these sampling requirements. As shown in Sect. 4.2, the model-assisted  $S_a$  will be relied upon in these regions. Also note the band of few retrievals around -160° longitude due to a collocated CALIOP/MODIS sampling artifact, which has been found in other studies (e.g., Ryan et al., 2024). The 1° x 1° latitude/longitude grid spacing makes this feature more pronounced.

We note that augmenting the MODIS AODs with AODs from the CALIPSO ODCOD retrievals (Ryan et al., 2024) would help increase our S<sub>a</sub> sample numbers, especially in polar regions. However, we chose instead to reserve the ODCOD dataset for an independent validation of the V5 AODs retrieved using the temporally and spatially varying S<sub>a</sub> reported in the newly developed S<sub>a</sub> tables (Sect. 4.3).

A pattern in  $S_a$  is evident (Fig. 3a), as larger  $S_a$  (> 40 sr) tend to be found near land masses, and smaller  $S_a$  (< 30 sr) are generally observed in the remote oceans (global median value of ~23

sr and global mean of ~25 sr; Table 3). This pattern in S<sub>a</sub> suggests different aerosol types/mixtures dominating in different regions. Larger S<sub>a</sub> indicates a mixture of marine and non-marine aerosols whereas smaller S<sub>a</sub> indicates more pristine "clean" marine aerosols. However, there are some regions in which S<sub>a</sub> are not enhanced near coasts (e.g., North America, western Europe, some of Africa) even though continental outflow exists in these regions. When long range aerosol transport and mixing into the MBL occurs at these locations, CALIOP may be identifying other aerosol types and the potentially impacted MBLs are being excluded.

In the remote oceans, S<sub>a</sub> varies with latitude. For example, remote oceanic S<sub>a</sub> in the Tropical region (about -20° to 20° latitude) are in the range of 25-40 sr, while those in the mid-to-high latitudes (< -20° or > 20°) are generally below 25 sr. This may be related to patterns in dimethyl sulfide (DMS) and/or chlorophyll over the oceans (e.g., Kettle et al., 1999), long-range transport of continental aerosols, or small biases in the MODIS retrieval. The S<sub>a</sub> patterns closely match those of Aqua MODIS AOD, and thus the higher AODs in the tropics may be influenced by a small AOD bias and/or the presence of non-sea salt aerosols. Also, it is possible there may be some stratospheric AOD biases in the CALIPSO L3 stratospheric aerosol product. The exact cause of this phenomenon is out of the scope of this paper, however, and thus is left for a separate study.

**Table 3.** June – August 2018 annual descriptive statistics for the global over-ocean non-gridded dataset of MODIS AOD constrained  $S_a$  for marine aerosols, only for those CALIOP aerosol profiles with some 5 km horizontal averaging and Feature Type QA = 3. These represent the points that were used to create Fig. 3.

|                    | Annual    |
|--------------------|-----------|
| Number             | 3,283,795 |
| Minimum            | 0.003 sr  |
| Maximum            | 145.01 sr |
| Mean               | 24.61 sr  |
| Median             | 23.37 sr  |
| Standard Deviation | 10.80 sr  |

A comparison of the S<sub>a</sub> literature review (Table 1) and Fig. 3a reveals there is a general agreement between the patterns of CALIOP-MODIS S<sub>a</sub> and the over-ocean S<sub>a</sub> in other studies obtained from a variety of methods/techniques. For example, the 36 sr and 33 sr in the southeast

Atlantic Ocean and Indian Ocean, respectively, agree well with the 30-40 sr range we find from our constrained S<sub>a</sub> retrievals. Also, the 34 sr value off the Asia coast is near our 35-45 sr constrained S<sub>a</sub>. In addition, the 23 sr value off the coast of southern Africa, indicative of a cleaner marine aerosol environment, agrees well with our values of less than 25 sr.

**Figure 4.** Twelve-year (2006-2018) spatial mean SSVF (Z< 2.5 km) from GEOS/GOCART at 1° x 1° latitude/longitude resolution (collocated with the constrained S<sub>a</sub> retrievals of Fig. 3a).




The twelve-year mean GOCART SSVF, collocated with the retrieved  $S_a$  (Fig. 3a), are shown in Fig. 4. These SSVF exclude dust and represent the total SS (i.e., fine and coarse mode SS aerosols). Smaller SSVFs (< 60%) are found near land masses, indicating the presence of advected pollution and/or biomass burning smoke aerosols in these regions. Conversely, in the remote oceans, the model SSVFs are large (> 90%) and suggest the presence of greater amounts of "pure" marine aerosols and thus less influence from pollution/biomass burning smoke. Note that this pattern is the inverse of the  $S_a$  spatial distribution (Fig. 3a), such that regions with low SSVFs generally correspond to higher  $S_a$ , and regions with high SSVFs generally correspond to lower  $S_a$ . Also, these spatial variations in  $S_a$  and SSVF are supported by patterns in MODIS fine mode fraction (FMF; not shown), with smaller FMFs found in the remote oceans and larger FMFs found near coasts, consistent with other MODIS FMF studies (e.g., Reid et al., 2022).

Sa gri
(Fi
Co
SS mo




As the next step, we quantify the relationship between modeled SSVF and the constrained  $S_a$  by computing the median constrained  $S_a$  as a function of SSVF (in 5% SSVF bins) using the gridded datasets of each parameter (Figs. 3a and 4). Figure 5 shows MODIS AOD constrained  $S_a$  (Fig. 3a) binned as a function of modeled SSVF (Fig. 4) in a series of box and whisker plots. Consistent with the spatial patterns discussed previously, there is a distinct increase in  $S_a$  as the SSVF decreases. This is due to other types of aerosols (e.g., anthropogenic pollution) becoming more dominant than sea salt aerosols when SSVF is low. A  $2^{nd}$  order polynomial fit to the medians of these data (Eq. 1) yields model-assisted  $S_a$  ( $S_{a,model}$ ) intercept values of ~21 sr for SSVF of 100% (i.e., "pure" marine) and ~58 sr for SSVF of 0% (i.e., no marine aerosols present). Figure 5 also shows the number of 1° x 1° latitude/longitude grid boxes in each 5% SSVF bin. The number of points per bin increase with increasing SSVF, ranging from 17 for the 0-5% SSVF bin to over 14,000 for the 95-100% SSVF bin.

$$S_{a, \text{model}} = 57.5 - 33.4(SSVF) - 3.2(SSVF^2)$$
 (1)

**Figure 5.** Box and whisker plots of median MODIS AOD constrained S<sub>a</sub> retrievals as a function of collocated modeled SSVF (binned for every 5% SSVF). The whiskers show the minimum and maximum values of each bin, and the boxplot notches indicate the confidence intervals around the median for each box. The red curve denotes the second order polynomial fit to the medians of each boxplot, with intersect values of 57.5 sr for a SSVF of 0% and 20.9 sr for a SSVF of 100%.

The light blue bars show the number of points (i.e., in 1° x 1° latitude/longitude grid boxes) per SSVF bin.

## 

## 4.2 Creating the seasonal Sa climatologies (2° x 4.8° latitude/longitude grid)

In the previous section, we discussed the details of establishing the relationship between the annual modeled SSVFs and Sa retrievals using data aggregated on a 1° x 1° grid, as this is the native resolution of the GEOS GOCART simulations used here. However, after conducting a CALIOP sampling analysis that considers the 16-day CALIPSO orbit repeat cycle (not shown), we found 2° x 4.8° is the optimal grid spacing to maximize the uniformity of CALIOP samples per latitude/longitude bin while still maintaining the regional fidelity of the lidar dataset. Thus, from this point forward, all maps shown in this paper will be shown at 2° x 4.8° latitude/longitude resolution. Additionally, as discussed earlier, the goal of this study is to establish CALIOP-classified marine Sa maps on seasonal scales. The analyses were thus segmented into four seasons: December, January, and February (DJF), March, April, and May (MAM), June, July, and August (JJA), and September, October, and November (SON). In this section, we describe the process and results of building the CALIPSO V5 CALIOP-classified marine aerosol Sa maps on seasonal scales using the modeled SSVF/ Sa retrieval relationship from Sect. 4.1.

The process begins with seasonal maps of the median S<sub>a</sub> from retrievals alone, as shown in Fig. 6. Here we require a minimum of 50 points in each latitude/longitude grid box for each season to compute the median S<sub>a</sub> value. This threshold was selected after conducting sensitivity studies to ensure a statistically robust characterization of the S<sub>a</sub>, while also accounting for satellite data coverage seasonally within each grid box over the study period. Compared to the annual S<sub>a</sub> retrieval map (Fig. 3a), the seasonal retrieval counts in Fig. 6 exhibit sometimes large decreases that vary by season. This is most notable in the Northern Hemisphere (NH) during JJA (Fig. 6c) but also occurs in the Southern Oceans and Arctic region. The lack of data in the NH is due to sun glint from MODIS that happens in the months of June and July (e.g., Kittaka et al., 2011), which results in few AOD retrievals and thus few constrained S<sub>a</sub> retrievals. Also, note the lack of retrievals over the waters surrounding the Indian Subcontinent in MAM (possibly due to cloud cover) and Oceania for each season (possibly due to significant cloud cover associated with the Indo-Pacific Warm Pool). For context, the number of samples for each grid box meeting our 50-

point minimum requirement is shown in Fig. 7, with areas of greatest sampling in the remote Pacific Ocean and southern Indian Ocean. In terms of S<sub>a</sub> value, the seasonal S<sub>a</sub> retrievals show a pattern similar to the twelve-year median S<sub>a</sub> (Fig. 3a) for most seasons, with higher S<sub>a</sub> in the Tropics and lower in mid to high latitudes. Also, elevated S<sub>a</sub> values are evident in the Bay of Bengal and Arabian Sea in DJF (Fig. 6a) and SON (Fig. 6d).

**Figure 6.** Twelve-year (2006-2018) spatial median of MODIS AOD constrained S<sub>a</sub> retrievals at 2° x 4.8° latitude/longitude resolution during daytime for CALIOP-classified marine aerosols for (a) DJF, (b) MAM, (c) JJA, and (d) SON. Medians are shown for those grid boxes containing at least 50 points.

The next step is to create maps of mean modeled SSVF at 2° x 4.8° grid spacing by regridding the 1° x 1° SSVFs to this coarser resolution using the twelve-year (2006-2018) GEOS/GOCART dataset (i.e., averaging all of the 1° x 1° SSVFs that are found within each 2° x 4.8° grid box). The resultant mean SSVFs below 2.5 km for each season are shown in Fig. 8. For all seasons, large SSVFs (> 90%) are found for most of the oceans (especially in remote regions), while lower SSVFs are found near coastlines and in the Arctic. For the Bay of Bengal and Arabian Sea, lower SSVFs are found for all seasons except JJA. These patterns are indicative of seasonal

aerosol transport based on the global atmospheric circulation simulated by the GOCART model, including the Indian monsoon (as discussed in more detail in Sect. 4.4). The Southern Oceans exhibit a decrease in SSVF compared to other remote ocean regions, but this is not nearly as pronounced as in the Arctic, for which low SSVFs are found (e.g., 

Figure 7. Twelve-year (2006-2018) number of samples per 2° x 4.8° latitude/longitude grid box of MODIS AOD constrained S<sub>a</sub> retrievals during daytime for CALIOP-classified marine aerosols, only for those grid boxes with at least 50 points (Fig. 6), for (a) DJF, (b) MAM, (c) JJA, and (d) SON.

**Figure 8.** Twelve-year (2006-2018) spatial mean SSVF (Z< 2.5 km) from GEOS/GOCART at 2° x 4.8° latitude/longitude resolution for (a) DJF, (b) MAM, (c) JJA, and (d) SON.

The maps of Fig. 8 are next used with Eqn. 1 to create the model-assisted  $S_a$  maps shown in Fig. 9. Clear patterns of  $S_a$  are found, with lower  $S_a$  in areas of high SSVF (e.g., remote oceans) and higher  $S_a$  in areas of low SSVF (e.g., near coasts). The  $S_a$  values in Fig. 9 range from ~21 sr to ~58 sr, as these are the intersect values of Eqn. 1. A region with some of the highest model-assisted  $S_a$  is the Arctic, for which low SSVFs are found (Fig. 8). This is most pronounced in MAM (Fig. 9b) and JJA (Fig. 9c). These large (> 50 sr) model-assisted  $S_a$  are consistent with relatively small sample of 532 nm Raman lidar observations in the Arctic. For example,  $S_a$  up to

 $\sim$ 50 sr were found during the spring 2014 Arctic haze season in Spitzbergen (Ritter et al., 2016), and even larger  $S_a$  (58-82 sr) were measured in this same region during an Arctic haze event the following spring (Stachlewska et al., 2018). In addition, Engelmann et al. (2021) observed  $S_a$  greater than 70 sr in the North Pole region (85-88.5° N), which they attribute to long-range transport of smoke aerosols.

**Figure 9.** Twelve-year (2006-2018) model-assisted S<sub>a</sub> derived using Fig. 8 and Eqn. 1 at 2° x 4.8° latitude/longitude resolution for (a) DJF, (b) MAM, (c) JJA, and (d) SON.

The benefit of Fig. 9 is that we now have global coverage (i.e., a strength of this model approach) of S<sub>a</sub>, whereas the empirically derived S<sub>a</sub> coverage is lacking in some areas. However, the intended purpose of these model-assisted maps is not to replace the retrievals, but to fill in the regions where there are no retrievals. Thus, we merged the seasonal S<sub>a</sub> maps of Fig. 6 and Fig. 9 to create "hybrid" retrieval/model-assisted maps, for which each 2° x 4.8° grid box includes either:

a) a S<sub>a</sub> retrieval if available and meets the 50-point minimum requirement or b) a model-assisted S<sub>a</sub> value for all other grid boxes. However, we implemented two additional procedures in creating

the final V5 marine  $S_a$  maps. For one, based on the field measurements shown in Table 2, we set a default minimum  $S_a$  value of 15 sr (i.e., if median  $S_a$  value is less than 15 sr, we set it to 15 sr). Secondly, we implemented an outlier replacement procedure that replaced outliers with the median of the surrounding 8 grid boxes (whether retrieved or model-assisted) whenever the absolute value of the relative difference of the  $S_a$  in the center pixel of a 3 × 3 grid was 30% greater than the median of the surrounding grid boxes. This was done to address some significant discontinuities observed in earlier test versions of the  $S_a$  maps. However, they only accounted for ~1-2% of all grid boxes over water (Fig. 12).

**Figure 10.** Twelve-year (2006-2018) hybrid  $S_a$  for CALIOP-classified marine aerosols, using the constrained retrieval, model estimation, default minimum, and outlier replacement methods, at  $2^{\circ}$  x 4.8° latitude/longitude resolution for (a) DJF, (b) MAM, (c) JJA, and (d) SON. These maps represent the marine  $S_a$  tables used to create the CALIPSO Version 5 (V5) data products.

Figure 10 shows the resultant final V5 S<sub>a</sub> maps for each season for CALIPSO-classified marine aerosols. Wide areas of the oceans are characterized by S<sub>a</sub> less than 25 sr, with some regions less than 20 sr (e.g., Southern Oceans, especially in MAM and JJA). S<sub>a</sub> increase south of ~60°S latitude, especially in the DJF season. The largest S<sub>a</sub> (> 50 sr) are found in the coastal regions, including Bay of Bengal, Arabian Sea, off the coast of Asia, west coast of Africa, and the Arctic region. While the minimum S<sub>a</sub> is forced to 15 sr for all seasons, the maximum S<sub>a</sub> value is ~56 sr for MAM. This is a model-derived S<sub>a</sub> in the Bohai Sea (near China) that corresponds to a SSVF of 3.5%. The maximum value for JJA is modeled as ~57 sr, located in the Caspian Sea (Middle East) and corresponding to a SSVF of 2.5%. The maximum values for SON and DJF (both ~63 sr) are retrievals near the coast in the northern Bay of Bengal and thus are not influenced by modeled SSVF. The magnitudes of these values are undoubtedly influenced by high concentrations of anthropogenic aerosols in the region.

**Figure 11.** Twelve-year (2006-2018) S<sub>a</sub> relative uncertainties for CALIOP-classified marine aerosols at 2° x 4.8° latitude/longitude resolution for (a) DJF, (b) MAM, (c) JJA, and (d) SON.

Each  $S_a$  of Fig. 10 is assigned a relative uncertainty value based on the following procedure. For those grid boxes with  $S_a$  retrievals, the uncertainty is computed as the median absolute deviation (MAD) divided by the median. This value is used provided it is not greater than the default V4 CALIPSO marine aerosol  $S_a$  relative uncertainty of 22% (Kim et al., 2018). If it is greater, it is set to 22%. Likewise, those grid boxes that use the model-assisted  $S_a$  or are assigned the default minimum value of 15 sr are also assigned a relative uncertainty of 22%. The resultant  $S_a$  relative uncertainty seasonal maps are shown in Fig. 11. Areas in red indicate those grid boxes with highest uncertainties (i.e., 22%), whereas regions for which there are retrievals available generally exhibit uncertainties between 10 and 20%. Note that for those grid cells with retrievals and an assigned uncertainty of 22%, the uncertainty median  $\pm$  uncertainty MAD prior to assignment is 25%  $\pm$  2% (DJF and MAM) and 26%  $\pm$  2% (JJA and SON).

Figure 12 illustrates the method used to obtain the S<sub>a</sub> value of each grid box for each season (Fig. 10). Those grid boxes with retrievals are shown in black and generally dominate the maps (with the exception of JJA). Model-assisted S<sub>a</sub> are denoted in red, and include regions such as the Southern Oceans, Arctic, and Indonesia during all seasons, most of the Northern Hemisphere during JJA, and the Bay of Bengal/Arabian Sea during MAM. Grid boxes colored green denote the default minimum S<sub>a</sub> value of 15 sr was used, including in DJF (North Atlantic), MAM and JJA (Southern Oceans), and SON (a few isolated grid boxes in the Southern Oceans and North Atlantic). Finally, outlier S<sub>a</sub> computed from the smoothing procedure are shown in blue. While outliers are infrequent and located in various regions across the global oceans, they are mostly situated at the default minimum-to-model boundary around ~60° S in JJA (Fig. 12c).

Figure 12. The S<sub>a</sub> method flag denoting the method used to obtain the twelve-year (2006-2018) hybrid S<sub>a</sub> shown in Fig. 11, consisting of either constrained retrieval (black), model estimation (red), default minimum (green), or outlier replacement (blue). These are provided at 2° x 4.8° latitude/longitude resolution for (a) DJF, (b) MAM, (c) JJA, and (d) SON.

## 4.3 Differences between V4.51 and V5 CALIPSO aerosol extinction and AOD, and preliminary validation study with ODCOD

650

655

Now that we have updated S<sub>a</sub> values for marine aerosols as a function of region and season, we can assess the impact these S<sub>a</sub> values have on CALIPSO L2 aerosol extinction and AOD retrievals. Note, however, that our intent is limited to providing a preliminary analysis, as the purpose of this paper is to document our technique and provide updates of the V5 CALIPSO S<sub>a</sub> to the community, as opposed to large-scale validation (a topic planned for a future paper). The seasonal S<sub>a</sub> maps (Fig. 10) were used in a V5 prerelease of the CALIPSO data processing software to retrieve new aerosol extinction profiles and tropospheric AODs. Four months (January, April, July, and October) of 2015 were chosen for this analysis, to ensure one month from each season

was represented. We report the differences in aerosol extinction coefficients and mean AOD between V4.51 and those from the V5 prerelease (V5-PR) data. We also use the AOD computed from the CALIPSO ODCOD algorithm as an independent source of validation, as it provides an estimate of total column optical depth retrieved from the CALIOP backscatter signal return of the ocean surface (Ryan et al., 2024). ODCOD is compared with both the standard V4.51 CALIPSO tropospheric AOD and the CALIPSO V5-PR AOD obtained using the revised S<sub>a</sub> developed in this work (Fig. 10). Note that the V4.51 ODCOD dataset has been validated against coastal/island AERONET AODs with a near-zero bias (0.011) and a root-mean-square-error (RMSE) of 0.12 (60%) (Thorsen et al., 2025).

Specifically, daytime and nighttime granules of the CAL LID L2 05kmAPro-Standard-V4-51 and CAL LID L2 05kmAPro-Standard-V5-00-PR products were leveraged during this of V5-PR aerosol extinction analysis coefficients and **AODs** through "Extinction Coefficient 532" "Column Optical Depth Tropospheric Aerosols 532" and The of parameters. **AODs** were compared against those "ODCOD Effective Optical Depth 532" parameter found in the CAL LID L2 05kmMLay-Standard-V5-00 product. The "Scene Flag" in this product was used to ensure the use of only cloud-free profiles containing only CALIOP-classified marine aerosols. For a more robust analysis, we also filter these data for only those ODCOD retrievals for which Bit 7 of "ODCOD QC Flag 532" is not set, thus indicating a confident retrieval (Ryan et al., 2024). These confident retrievals require all of the following conditions be met: the ODCOD Effective Optical Depth 532 retrieval must be valid (i.e., not -9999.0), all single shots of the averaged L1 attenuated backscatter profile must have the same number of bins shifted (i.e., the "ssNumber Bins Shift" parameter in the CAL LID L2 05kmMLay-Standard-V5-00 product), the AMSR corrected MERRA-2 wind speed (i.e., magnitude of the reported ODCOD Surface Wind Speeds 10m plus the ODCOD Surface Wind Speed Correction) must be between 3 and 15 ms<sup>-1</sup>, the surface integrated depolarization ratio (SIDR) must be less than or equal to 0.05, and the surface 532 nm integrated attenuated backscatter (SIAB) must be less than or equal to 0.0413 sr<sup>-1</sup> (daytime) or less than or equal to 0.0353 sr<sup>-1</sup> (nighttime). This procedure provides a strictly filtered and robust subsample of all over-ocean cloud-free profiles that are used in our preliminary V5-PR analysis.

The aerosol extinction coefficients from V4.51 and V5-PR, and mean AODs from V4.51, V5-PR, and ODCOD, are compared for each of the four months (January, April, July, and October of 2015) for Global Oceans and seven regions: Southern Oceans (R1), Bay of Bengal and Arabian Sea (R2), Remote Pacific Ocean (R3), North Atlantic Ocean (R4), West Coast of North America (R5), Asia Coast (R6), and West Coast of Africa (R7). The latitude and longitude boundaries for each region are shown spatially in Fig. 13. While some regions encompass a large amount of land, only the oceanic parts of each domain are used in the analysis. These regions were selected specifically to capture different aerosol model scenarios, including coastal (typically low SSVF, thus higher Sa) and open oceans (typically high SSVF, thus lower Sa), and various derived-Sa regimes in general (e.g., model versus retrieval).

**Figure 13.** The latitude and longitude boundaries for each of the seven regions of the aerosol extinction coefficient and AOD study (Sect. 4.3), including Southern Oceans (**R1**; -90° to -50°, -180° to 180°), Bay of Bengal and Arabian Sea (**R2**; 10° to 25°, 60° to 95°), Remote Pacific Ocean (**R3**; -15° to 5°, -175° to -105°), North Atlantic Ocean (**R4**; 35° to 90°, -60° to 0°), West Coast of North America (**R5**; 25° to 50°, -128° to -110°), Asia Coast (**R6**; 20° to 55°, 110° to 140°), and West Coast of Africa (**R7**; -25° to 15°, -15° to 15°).

Figure 14 shows examples of the daytime comparisons of V4.51 and V5-PR CALIPSO aerosol extinction coefficient retrievals for only those profiles with CALIOP-classified marine aerosols (as determined by the L2 CALIPSO VFM product) for two regions (Southern Oceans and Bay of Bengal/Arabian Sea) and two months (January and July 2015). For context, the

corresponding S<sub>a</sub> differences are shown in the histograms of Fig. 15, computed using the "Initial\_Lidar\_Ratio\_Aerosols\_532" parameter in the CAL\_LID\_L2\_05kmALay products as V5-PR – V4.51 (i.e., V5-PR – 23 sr). For the Southern Oceans during January 2015 (Fig. 14a), most points fall along the one-to-one line and thus indicate little change in aerosol extinction between V4.51 and V5-PR in this region and season (i.e., little departure between the V5-PR S<sub>a</sub>, as shown in Fig. 10a, and the fixed V4.51 S<sub>a</sub> value of 23 sr). A near-zero (0.44 sr) mean difference in V5-PR-V4.51 initial S<sub>a</sub> is found for this region/month (Fig. 15a). However, in July 2015 (Fig. 14c), lower aerosol extinction retrievals are found for V5-PR compared to V4.51, as a result of S<sub>a</sub> lower than 23 sr (Fig. 10c; with a mean difference of -3.59 sr, as shown in Fig. 15c).

In the Bay of Bengal/Arabian Sea region, the V5-PR aerosol extinction coefficients are far larger than those from V4.51 during January 2015 (Fig. 14b), resulting from the much larger V5-PR S<sub>a</sub> used in this region and season (Fig. 10a) compared to 23 sr (mean difference of 29.34 sr, as shown in Fig. 15b). The V5-PR S<sub>a</sub> are smaller during JJA (Fig. 10c) and thus the resultant V5-PR aerosol extinction coefficients for July 2015 are closer in agreement with those from V4.51 yet still a bit larger (Fig. 14d). The corresponding mean S<sub>a</sub> difference is 5.68 sr (Fig. 15d). This region is discussed further in a case study in Sect. 4.4.

**Figure 14.** Scatterplots of daytime 532 nm Level 2 (L2) aerosol extinction coefficient retrievals for CALIOP-classified marine aerosols from the V4.51 versus V5-PRCALIPSO data products for the Southern Oceans region (-90° to -50°, -180° to 180°) during (a) January 2015 and (c) July 2015, as well as the Bay of Bengal and Arabian Sea region (10° to 25° N latitude, 60° to 95° E longitude) during (b) January 2015 and (d) July 2015. The scatterplots are color-coded by number density and the black line is the one-to-one line.

**Figure 15.** Histograms of daytime 532 nm Level 2 (L2) initial S<sub>a</sub> differences between the V4.51 and V5-PR CALIPSO data products (V5-PR – V4.51) for CALIOP-classified marine aerosols for the Southern Oceans region (-90° to -50°, -180° to 180°) during (a) January 2015 and (c) July 2015, as well as the Bay of Bengal and Arabian Sea region (10° to 25° N latitude, 60° to 95° E longitude) during (b) January 2015 and (d) July 2015.

The results of the daytime AOD analysis for four months of 2015 (January, April, July, and October) are shown in the bar plots of Fig. 16, with mean V4.51 AOD (in blue), mean V5-PR AOD (in orange), and mean ODCOD (in yellow). Globally, V5-PR AODs are larger than V4.51, but only by a small amount (i.e., ~0.01-0.02). Similarly for most regions/seasons, V5-PR AODs are larger than V4.51. This is indicative of larger V5-PR S<sub>a</sub> in those regions/seasons compared to V4.51 value of 23 sr. Sometimes the increase in AOD from V4.51 to V5-PR is minimal (e.g., ~0.01 in the Remote Pacific in October 2015; R3 in Fig. 16d). However, the region with the largest changes in AOD is the Bay of Bengal and Arabian Sea (R2), particularly during January 2015, with an AOD increase of ~0.20 (Fig. 16a). This is indicative of a much larger V5-PR S<sub>a</sub> compared to V4.51 (as examined in the case study of Sect. 4.4). For other regions, like the Southern Oceans (R1), the V5-PR AOD is consistently the same or lower than 4.51, a direct result of using a S<sub>a</sub> value similar or lower than 23 sr in this area.

The differences between V4.51 AOD and ODCOD (Fig. 16) demonstrate the performance of the V4.51 standard retrieval relative to ODCOD (our "truth" dataset) and quantify the deficiencies in the ability of the standard V4.51 CALIOP retrieval to reliably estimate the column AOD. These deficiencies can be due to both Sa selection and layer detection, such that even if the correct Sa is used, the standard retrieval is expected to be lower than ODCOD. This can be attributed to optically thin layers that are below CALIOP's direct detection thresholds and are not detected as features in the standard retrieval but are responsible for attenuation that is accounted for in the ODCOD retrieval. Toth et al. (2018) suggests that the standard retrieval generally fails to detect any layers when the column optical depths are below ~0.06 (estimated globally, not regionally).







Globally and for most regions/seasons, ODCOD is greater than V4.51 (as expected, i.e., due at least partly to layer detection), most notably in the Bay of Bengal and Arabian Sea during January and April 2015. The differences between the V5-PR AOD and ODCOD demonstrate the performance of the seasonally and regionally varying Sa maps relative to ODCOD, and these are found to be generally smaller than the V4.51-ODCOD differences (i.e., V5-PR AODs exhibit a better agreement with ODCOD than V4.51, as expected). For example, in the Bay of Bengal/Arabian Sea during January 2015, the difference in mean AOD changes from ~0.24 between ODCOD and V4.51 to ~0.04 between ODCOD and V5-PR (note that the 0.04 value is comparable to the 0.06 value reported in Toth et al. 2018). This scenario illustrates the improvements to CALIOP AOD due to the use of the new Sa maps versus a fixed value of Sa for marine aerosols. However, differences in mean AOD (> ~0.02-0.03) still exist between V5-PR AOD and ODCOD for the global oceans (and larger for some regions/seasons), even after implementing our regionally and seasonally varying S<sub>a</sub> (e.g., the ODCOD vs. V5-PR difference of ~0.19 for the Bay of Bengal/Arabian Sea in April 2015). Again, these are likely due to detection deficiencies in the standard CALIOP aerosol retrieval that are not an issue for the ODCOD algorithm (Ryan et al., 2024).

Note that results similar to those shown in Fig. 16 are found for a nighttime analysis, provided as bar plots in the appendix (Fig. A1). Also, for context, we include in the appendix daytime bar plots for those 5 km CALIOP segments in which collocated Aqua MODIS AODs are available in addition to V4.51, V5-PR, and ODCOD (Fig. A2; however, this analysis is not as robust due to the relatively low number of MODIS data points for several seasons/regions). As a

final remark, we note that uncertainties exist in the ODCOD and standard AOD retrievals. For example, Ryan et al. (2024) report a global ODCOD median random uncertainty of  $\sim$ 0.11  $\pm$  0.01. Thus, the statistical robustness of the comparisons likely varies as a function of month/region.




**Figure 16.** Bar plots of daytime mean aerosol optical depth (AOD) for CALIPSO Version 4.51 (V4.51; in blue), Version 5 (V5-PR; in orange), and ODCOD (in yellow) for (a) January 2015, (b) April 2015, (c) July 2015, and (d) October 2015. Mean AODs are shown for Global Oceans and for seven regions: Southern Oceans (R1; -90° to -50°, -180° to 180°), Bay of Bengal and Arabian Sea (R2; 10° to 25°, 60° to 95°), Remote Pacific Ocean (R3; -15° to 5°, -175° to -105°), North Atlantic Ocean (R4; 35° to 90°, -60° to 0°), West Coast of North America (R5; 25° to 50°, -128° to -110°), Asia Coast (R6; 20° to 55°, 110° to 140°), and West Coast of Africa (R7; -25° to 15°, -15° to 15°). These analyses are subsampled for those CALIOP 5 km segments with valid retrievals of V4.51 tropospheric AOD, V5-PR tropospheric AOD, and ODCOD, are cloud-free, and contain only marine aerosols.

## 4.4 Sa case study: Bay of Bengal and Arabian Sea

As discussed in Sect. 4.3, the Bay of Bengal and Arabian Sea region featured the greatest changes to L2 CALIPSO tropospheric AOD (specifically in January 2015) when using the new seasonal S<sub>a</sub> maps to retrieve aerosol extinction rather than a fixed S<sub>a</sub> value. However, this was not the case in July 2015, as a much smaller change in AOD was found for this region (Fig. 16c). In this section, we explore this seasonality and link it to seasonal changes in wind speed magnitude and direction due to Indian monsoon patterns.

Fig. 17a shows the 2006-2018 spatial mean modeled SSVF below 2.5 km for the DJF season, with low SSVFs (below 65%) for the entire region. This is consistent with the generally low wind speeds and northeast wind flow found during DJF in the Bay of Bengal and Arabian Sea (e.g., Shankar et al., 2002; Wang et al., 2020). Wind speed impacts the production of sea salt aerosols and is highly influential in modeling the amount of sea salt aerosols, as models parameterize sea salt emissions by wind speed (Chin et al., 2002). Lower wind speeds result in less sea salt aerosol, so, with all else being equal, would produce lower SSVFs. As for wind flow, since the prevailing pattern is from the northeast due to the Winter Indian Monsoon, there is a greater opportunity for transport of smoke/pollution from land sources into the marine environment and thus also lower the SSVFs. These patterns are consistent with the DJF Sa map (Fig. 17c), as much of the region exhibits Sa of greater than 45 sr, indicating a pollution/marine aerosol mixture. The opposite patterns are found for the JJA season, with larger SSVFs (Fig. 17b) and smaller Sa (Fig. 17d). This is consistent with greater wind speeds (i.e., more sea salt production) and prevailing southwest flow due to the Summer Indian Monsoon (i.e., less pollution transport).

**Figure 17.** For 2006-2018 at  $2^{\circ}$  x 4.8° latitude/longitude resolution, spatial mean SSVF (Z < 2.5 km) from GEOS/GOCART for (a) DJF and (b) JJA, and hybrid  $S_a$  map from constrained retrievals and model estimations for (c) DJF and (d) JJA, for the Bay of Bengal and Arabian Sea region ( $10^{\circ}$  to  $25^{\circ}$  N latitude,  $60^{\circ}$  to  $95^{\circ}$  E longitude).

Figure 18 shows the evaluation of the tropospheric CALIPSO AODs in the Bay of Bengal/Arabian Sea region due to the new S<sub>a</sub> (V5-PR), shown here in 2D histogram form as an extension of the analyses from Sect. 4.3. Figure 18b reveals a better agreement between ODCOD and the V5-PR CALIOP AOD (slope=0.69) than between ODCOD and the V4.51 standard CALIOP AOD (slope=0.14; Fig. 18a). The RMSE also decreases for the ODCOD/V5-PR CALIOP AOD analysis (0.19; Fig. 18b) compared to that of ODCOD/V4.51 standard CALIOP AOD (0.27; Fig. 18a). This improvement in January 2015 is a result of the larger S<sub>a</sub> (mostly retrievals) used in this region and season (Fig. 17c) compared to the fixed V4.51 CALIPSO marine S<sub>a</sub> of 23 sr. Note that this is even more evident during comparisons to Aqua MODIS AOD (Fig.

A3). The results of this case study demonstrate the importance of performing these S<sub>a</sub> analyses on seasonal scales.

**Figure 18.** For the Bay of Bengal and Arabian Sea region (10° to 25° N latitude, 60° to 95° E longitude) during January 2015, 2D histograms of ODCOD against the (a) V4.51 CALIOP AOD and (b) V5-PR CALIOP AOD (i.e., using the seasonal and regional S<sub>a</sub>), all at 532 nm. The dashed lines indicate the one-to-one lines, and the solid black lines show the lines-of-best-fit.

# 4.5 Validation using ground-based AOD retrievals from AERONET






In the previous section, we evaluated the differences in AOD between CALIPSO Version 4.51 (fixed S<sub>a</sub>) and the V5-PR AODs (S<sub>a</sub> tables) and the relationship between these AODs and ODCOD for a four-month period. Here, we perform a more extensive (June 2006-October 2022) validation of the V5-PR CALIPSO AODs using coastal and island AERONET measurements and contrasting that analysis with Version 4.51 AODs. NASA's AERONET is a global, ground-based sun photometer network that has been used for over three decades as the primary means for the validation of spaceborne aerosol retrievals (Holben et al., 1998). AOD retrievals from AERONET report uncertainties of ± 0.01-0.02 (Eck et al., 1999; Barreto et al., 2016; Giles et al., 2019). The approach taken here exactly follows the study of Thorsen et al. (2025). In brief, V3 L2 cloud-screened and quality-assured AODs (Giles et al., 2019) are used, after interpolation to 532 nm using a 2<sup>nd</sup> order polynomial fit (Eck et al., 1999; Schuster et al., 2006). These AERONET AODs from coastal and island sites are spatially (within 80 km) and temporally (within 2 hours) collocated with over-water CALIPSO profiles. Further methodology details (e.g., filtering, averaging, significance testing, etc.) can be found in Thorsen et al. (2025). Lastly, we limit the analysis to samples with at least one CALIOP layer is classified as marine aerosol, that is, other

aerosol types may also be included in the vertical profile. This methodological choice enables us to increase the data yield, allowing for a statistically robust analysis.

AOD comparisons between AERONET and CALIPSO are depicted in Fig. 19. From V4.51 to the V5-PR AODs, RMSE decreases from 0.16 (88%) to 0.13 (72%) and bias decreases from -0.049 (-28%) to -0.024 (-14%). Both V4.51 and V5-PR AODs exhibit significant (p 

**Figure 19.** 2D histograms of AERONET AOD against (a) CALIPSO Version 4.51 AOD and (b) CALIPSO Version 5 AOD for June 2006 through October 2022, with at least one marine aerosol layer present in the CALIPSO profiles.

Note that the validation efforts of the V5 S<sub>a</sub> in this paper focused on a column-integrated aerosol perspective (i.e., AOD and comparisons with ODCOD and AERONET). However, we carried out preliminary investigations of CALIPSO aerosol extinction profiles collocated with data from airborne High Spectral Resolution Lidar (HSRL) underflights of CALIPSO, and only minimal changes between V4.51 and V5 were found (thus not provided here). This is because the

majority of underflights were in areas (e.g., Sargasso Sea) with small changes in S<sub>a</sub> (i.e., the V5 S<sub>a</sub> were similar to 23 sr for marine). Airborne HSRL underflights of CALIPSO are not available for regions in which we expect the greatest impact to aerosol extinction profiles (e.g., regions where the largest AOD changes were found, like the Bay of Bengal and Arabian Sea).






## **5.** Conclusions

Twelve-years (2006-2018) of NASA CALIOP attenuated backscatter profiles, constrained by Aqua MODIS AOD, were used to derive extinction-to-backscatter ratios, known as lidar ratios (S<sub>a</sub>), over oceans during daytime conditions at 532 nm. The S<sub>a</sub> analysis was subsampled for only those CALIOP aerosol layers classified as "marine", as determined by the CALIOP aerosol typing algorithm. In an improvement over the current Version 4.51 (V4.51) S<sub>a</sub> selection scheme that assigns a single Sa per aerosol type per wavelength, this work focuses on the creation of regional and seasonal Sa tables (at 2° x 4.8° latitude/longitude grid spacing) that have been incorporated into the Version 5.00 (V5) CALIPSO data products release. The V4.51 value of 23 sr for CALIOPclassified marine aerosol was updated with Sa values that vary both regionally and seasonally. The bulk of the Sa tables were produced through climatological maps of Sa retrievals constrained by MODIS AOD, but data sparse regions use model-assisted values derived using the relationship between the constrained retrievals and GEOS GOCART modeled sea salt volume fractions (SSVFs). The hybrid (retrieval + model) S<sub>a</sub> maps were used in initial validation studies by ingesting them into the CALIOP algorithms to produce new Version 5.0 prerelease (V5-PR) CALIOP aerosol extinction profiles and tropospheric AODs. These were then compared against the standard V4.51 CALIOP tropospheric AODs, the CALIPSO ODCOD parameter, and groundbased AERONET AOD retrievals.

925 The major findings of this study are:

(1) An inverse relationship is found between the modeled SSVFs and the AOD constrained  $S_a$  of CALIOP-classified marine aerosols. In the remote oceans, larger SSVFs (> 95%) correspond to smaller  $S_a$  (< 30 sr), more indicative of "pure" sea salt aerosols. Near land masses, smaller SSVFs (< 65%) correspond to larger  $S_a$  (> 40 sr), indicating the influence of aerosols from land sources. A second order polynomial fit to these data yields values of 21 sr for 100% SSVF and 58 sr for 0% SSVF.

- (2) Hybrid (retrieval + model) S<sub>a</sub> tables (i.e., latitude by longitude by season) were created for December-February (DJF), March-May (MAM), June-August (JJA), and September-November (SON). These maps capture the regional and seasonal variability of S<sub>a</sub>, including the atmospheric patterns/movement of aerosols. For example, the monsoon patterns near India influence the amount of sea salt aerosols versus over-land aerosols and thus impact the S<sub>a</sub> found over the Bay of Bengal and Arabian Sea. A case study of this region demonstrated the impact of the seasonal S<sub>a</sub> for DJF, during which the constrained S<sub>a</sub> retrievals (> 45 sr) are substantially larger than that of the V4.51 CALIOP-classified marine value of 23 sr, thus resulting in correspondingly larger aerosol extinction and AOD retrievals in the V5 data products.
  - (3) Global analysis of the selection method used to obtain S<sub>a</sub> for any location shows that MODIS-constrained retrievals are used over large areas of the oceans for most seasons, with the exception being the Northern Hemisphere in JJA, where MODIS sun glint causes greatly increased reliance on the model-assisted values. The model estimation method is also used in the polar regions due to a lack of MODIS-constrained S<sub>a</sub> retrievals.

- (4) An initial comparison was made between daytime V4.51 and V5-PR CALIPSO aerosol extinction coefficients retrieved over oceans within seven climatologically varying regions for four months in 2015 (January, April, July, and October). Similar comparisons were conducted using V5-PR AODs and collocated ODCOD retrievals. V5-PR AODs are generally larger (and better agree with ODCOD) than V4.51 AODs, as the Sa tables yield values greater than the 23 sr used uniformly by V4.51 over vast parts of the oceans. Globally, this difference is slight (~0.01-0.02), but some regionality exists. For example, a region with little change or a slight decrease is the Southern Oceans (i.e., V5-PR Sa are similar to or smaller than 23 sr). A region with a large increase in AOD (e.g., ~0.20 during January 2015) is the Bay of Bengal and Arabian Sea due to the large Sa increasing the retrieved aerosol extinction and subsequent AOD.
- (5) In a comparison with ground-based retrievals from coastal and island AERONET sites, the transition from V4.51 AODs to V5-PR AODs yields a root-mean-square-error decrease from 0.16 (88%) to 0.13 (72%) and a corresponding bias decrease from -0.049 (-28%) to -0.024 (-14%). This represents a modest improvement in the V5 AODs from that of V4.51 dataset which can be attributed directly to the V5 S<sub>a</sub> tables for CALIOP-classified marine aerosols.

965 In this study, we develop a synergistic fusion of passive and active remote sensing measurements to build a collection of marine aerosol S<sub>a</sub> maps with values that vary as a function of region and season. In the CALIPSO V5 data products, the initial lidar ratios for all aerosol layers classified as marine by the CALIOP aerosol subtyping algorithm are interpolated in both time and space from these maps. These interpolated values are reported in the CALIOP V5 data 970 products, as is a flag value that identifies these retrievals as being based on the Sa maps. Applying this technique over the ocean allows for a more realistic ocean-to-land Sa transition in coastal regions. In the previous CALIPSO Sa approach, a large step change was seen in the aerosol Sa over land and over water. The regional Sa tables created in this study help mitigate this issue and provide a smoother, more physically realistic transition in values. Despite the challenges of 975 retrieving robust passive AODs over land surfaces, the methods presented here to develop S<sub>a</sub> tables from AOD-constrained retrievals for over-ocean CALIOP aerosol types can, in principle, be applied to those found over land (dust, polluted dust, polluted continental/smoke, elevated smoke, and clean continental). The active/passive retrieval + aerosol model combined approach of developing S<sub>a</sub> tables documented in this study can be adopted by future satellite missions flying 980 elastic backscatter lidars in tandem with collocated passive sensors.

### Appendix A.

Figure A1. Bar plots of nighttime mean aerosol optical depth (AOD) for CALIPSO Version 4.51 (V4.51; in blue), Version 5 (V5; in orange), and ODCOD (in yellow) for (a) January 2015, (b) April 2015, (c) July 2015, and (d) October 2015. Mean AODs are shown for Global Oceans and for seven regions: Southern Oceans (R1; -90° to -50°, -180° to 180°), Bay of Bengal and Arabian Sea (R2; 10° to 25°, 60° to 95°), Remote Pacific Ocean (R3; 15° to 5°, -175° to -105°), North Atlantic Ocean (R4; 35° to 90°, -60° to 0°), West Coast of North America (R5; 25° to 50°, -128° to -110°), Asia Coast (R6; 20° to 55°, 110° to 140°), and West Coast of Africa (R7; -25° to 15°, -15° to 15°). These analyses are subsampled for those CALIOP 5 km segments with valid retrievals of V4.51 tropospheric AOD, V5 tropospheric AOD, and ODCOD. Note the lack of data for R2 during July 2015 due to the ODCOD filtering scheme described in Sect. 4.3.

Figure A2. Bar plots of daytime mean aerosol optical depth (AOD) for CALIPSO Version 4.51 (V4.51; in blue), Version 5 (V5; in orange), ODCOD (in yellow), and collocated Aqua MODIS (in purple) for (a) January 2015, (b) April 2015, (c) July 2015, and (d) October 2015. Mean AODs are shown for Global Oceans and for seven regions: Southern Oceans (R1; -90° to -50°, -180° to 180°), Bay of Bengal and Arabian Sea (R2; 10° to 25°, 60° to 95°), Remote Pacific Ocean (R3; 15° to 5°, -175° to -105°), North Atlantic Ocean (R4; 35° to 90°, -60° to 0°), West Coast of North America (R5; 25° to 50°, -128° to -110°), Asia Coast (R6; 20° to 55°, 110° to 140°), and West Coast of Africa (R7; -25° to 15°, -15° to 15°). These analyses are subsampled for those CALIOP 5 km segments with valid retrievals of V4.51 tropospheric AOD, V5 tropospheric AOD, ODCOD, and collocated Aqua MODIS AOD. Note the lack of data for R2 and R5 during July 2015.


**Figure A3.** For January 2015 and the Bay of Bengal and Arabian Sea region (10° to 25° N latitude, 60° to 95° E longitude), 2D histograms of Aqua MODIS AOD against the (a) V4.51 CALIOP AOD and (b) V5 CALIOP AOD (i.e., using the seasonal and regional S<sub>a</sub>), all at 532 nm. The dashed lines indicate the one-to-one lines, and the solid black lines show the lines-of-best-fit.



Code availability:

The Collopak toolkit for collocating satellite observations is distributed by the Space Science and Engineering Center at the University of Wisconsin – Madison and publicly available at <a href="https://www.ssec.wisc.edu/~gregq/collopak/">https://www.ssec.wisc.edu/~gregq/collopak/</a>.


Data availability:

CALIPSO data are available from the NASA Langley Research Center Atmospheric Science Data Center (ASDC), including the Version 4.51:

CAL\_LID\_L1-Standard-V4-51 (<u>https://doi.org/10.5067/CALIOP/CALIPSO/CAL\_LID\_L1-Standard-V4-51</u>)


 $CAL\_LID\_L2\_05kmAPro-Standard-V4-51\\ (\underline{https://doi.org/10.5067/CALIOP/CALIPSO/CAL\_LID\_L2\_05kmAPro-Standard-V4-51}\ )$ 

CAL LID L2 VFM-Standard-V4-51

(https://doi.org/10.5067/CALIOP/CALIPSO/CAL LID L2 VFM-Standard-V4-51)

CAL\_LID\_L3\_Stratospheric\_APro-Standard-V1-00 (https://doi.org/10.5067/CALIOP/CALIPSO/LID\_L3\_STRATOSPHERIC\_APRO-STANDARD-V1-00)


MODIS data are available from the Level-1 and Atmospheric Archive & Distribution System Distributed Active Archive Center (LAADS DAAC), including the Collection 6.1 Aqua MODIS 1 km Geolocation files: MYD03.061 (http://dx.doi.org/10.5067/MODIS/MYD03.061)

- 1075 GEOS model data, including simulations of AeroCom Upper Troposphere Lower Stratosphere (UTLS) experiments (<a href="https://aerocom.met.no/experiments/UTLS/">https://aerocom.met.no/experiments/UTLS/</a>), are available from the NASA Center for Climate Simulation (NCCS) server.
- AERONET data, including the Version 3 Level 2 data product, are available at the NASA AERONET webpage (<a href="https://aeronet.gsfc.nasa.gov/new\_web/webtool\_aod\_v3.html">https://aeronet.gsfc.nasa.gov/new\_web/webtool\_aod\_v3.html</a>).

#### Author contribution:

Conceptualization: TDT, GLS, MC, RAF, AEG, JK, RAR, CRT, MAV, and EJW; Formal analysis: TDT, MBC, ZL, DP, SDR, and TJT; Investigation/Methodology: TDT, GLS, MBC, ZL, DP, and SDR; Software: TDT, MBC, ZL, DP, and SDR; Supervision: GLS, CRT, and MAV; Validation: TJT and JK; Visualization: TDT, GLS, JK, and TJT; Writing (original draft preparation): TDT. All authors contributed to the writing of the manuscript during the review and editing phase.

### Competing interests:

The authors declare that they have no conflict of interest.

### Acknowledgements:

This work was funded by the NASA CALIPSO Project. We thank the AERONET PIs and Co-Is and their staffs for establishing and maintaining the AERONET sites used in this investigation.

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
