# Peer review of "Mapping 532 nm Lidar Ratios for CALIPSO-Classified Marine Aerosols using MODIS AOD Constrained Retrievals and GOCART Model Simulations"

_EGUsphere, 2025_

## Author Comment (AC1)

The paper by Toth et al. describes the creation of regional and seasonal lidar ratio tables for CALIPSO's final data release (V5). The study focuses on marine aerosol and for the creation of the lidar ratio tables passive and active remote sensing measurements were used, in particular MODIS AOD constrained CALIOP backscatter profiles in addition to GOCART model simulations of sea salt volume fraction (SSVF). This is a great step beyond the single globally-constant lidar ratio value that was used in earlier versions.

In addition to the well described methodology, differences between V4.51 and the new V5 CALIPSO extinction and AOD were also presented together with a preliminary validation using the CALIPSO ODCOD algorithm for seven study regions by utilizing four months from 2015. Improvements due to the new seasonal maps are demonstrated in a case study, focusing on one of the aforementioned regions. Validation using AERONET retrievals was also performed and the results suggest that the lidar ratio tables for marine aerosol improve the AOD retrievals.

Overall, the paper is of good quality, structured and well-written and should be published after only a few minor revisions (listed below).

Response: We thank the reviewer for the feedback and constructive comments, which we believe have helped improve the quality of this paper.

Line 84: It would be worth mentioning a few of the campaigns used, in addition to a number of studies that revealed a bias induced by the marine aerosol type in the CALIPSO data.

Response: We named some of the campaigns used, and these are also included in Table 2. We interpret "bias induced by the marine aerosol type" as a low bias in CALIPSO AOD for marine aerosols compared to other instruments/retrievals (e.g., MODIS, SODA, and HSRL). The following text was added in the revised manuscript:

"These include the Second Aerosol Characterization Experiment (ACE 2; e.g., Ansmann, 2001), Indian Ocean Experiment (INDOEX; e.g., Welton et al., 2002), and airborne High Spectral Resolution Lidar (HSRL) underflights of CALIPSO (e.g., Rogers et al., 2014). Relevant details of these campaigns are found in Table 2. Note that several studies reported lower marine aerosol optical depths (AODs) for CALIPSO compared to Moderate Resolution Imaging Spectroradio-meter (MODIS; e.g., Oo and Holz, 2011), Synergized Optical Depth of Aerosols (SODA; e.g., Dawson et al., 2015), and HSRL (e.g., Rogers et al., 2014). These discrepancies were at least partly attributed to the assignment of incorrect $S_a$, including through possible aerosol misclassification."

Line 86-89: Similarly, as above, one should acknowledge previous studies that highlighted the necessity for the introduction of the dusty marine type.

Response: As noted in Kim et al. (2018), we have added three previous studies that demonstrate the mixing of dust and marine aerosol in the Atlantic Ocean. We have also included a sentence concerning the Burton et al. (2013) study that reports HSRL lidar ratio measurements more indicative of dust and marine aerosol than dust and smoke aerosol for CALIPSO V3 "polluted dust" aerosol layers.

The modified sentences are included in the revised manuscript as follows:

"This type was added to account for mixtures of marine and dust aerosol occurring over the oceans, especially Saharan dust during transport across the Atlantic Ocean (e.g., Liu et al., 2008; Groß et al., 2016; Kuciauskas et al., 2018). In V3, these features would typically be classified (incorrectly) as polluted dust, as airborne HSRL measurements of $S_a$ for CALIPSO "polluted dust" aerosol layers (~35 sr) suggest a mixture of dust and marine as opposed to that of dust and smoke (Burton et al., 2013)."

Section 2.1: While I really appreciate the fact that the authors provide the filenames of the products that they've used, I find it slightly interruptive for reading. The data availability section contains the filename already. Same applies for sect. 4.3 and the names of the variables used.

Response: Thank you for this comment. We prefer to keep the product filenames in these sentences as currently written, as they clearly provide the reader the details of which datasets were used in our analyses.

Line 322: Please cite the available software properly.

Response: We have added a "Code Availability" section and cited collopak as follows:

"The Collopak toolkit for collocating satellite observations is distributed by the Space Science and Engineering Center at the University of Wisconsin – Madison and publicly available at https://www.ssec.wisc.edu/~gregq/collopak/."

Lines 328-330: Why is the lidar ratio allowed to vary to physically meaningless values, i.e., -50 sr? Aren't the retrievals stable for a range 0 to 150 sr? Could the authors also provide references for the sensitivity studies that they mention?

Response: Yes, the retrievals were stable for the 0 to 150 sr range, but we chose -50 to 150 sr instead as the range for the Fernald iterations because we wanted to capture a wide spectrum of lidar ratios and examine the impact it had on the constrained retrievals. Our approach is similar to using a bisection method which establishes an initial zero crossing interval by choosing initial values of $S_a = -50$ sr and $S_a = 150$ sr. To emphasize what is stated in the paper, note that a very small (~0.05%) number of resultant lidar ratio retrievals were negative (and we use median values for our maps), thus the negative lidar ratios have a negligible impact on our results. Also, there are no references for the sensitivity studies we mentioned, as these were internal analyses we conducted as part of the work for this paper. We have modified the sentence in question to the following:

"$S_a$ are allowed to vary over a range from $-50$ sr to 150 sr to capture a wide spectrum of $S_a$ and because the iterations for the Fernald retrieval were numerically stable for this range (determined through sensitivity studies)."

Line 340: Please explain the quality flags selected for the MODIS data. A reader might not know what a Land Ocean Quality Flag value greater or equal to 1 means.

Response: The Land Ocean Quality Flag is interpreted as follows: 0=bad retrieval, 1=marginal, 2=good, and 3=very good. We have rewritten that sentence in the manuscript as follows:

"To ensure high quality Ångström interpolations we required positive values for all four MODIS AODs and rejected those cases flagged as "bad retrievals" by MODIS's Land Ocean Quality Flag."

Lines 361-368: It is not clear why the additional filtering step was used. Please explain and also summarize in 1-2 sentences the main points of Li et al., 2022 regarding the SNR/classification confidence relationship.

Response: This filtering step was used to increase the confidence of the CALIOP aerosol classification, which is important for this study because of our focus on CALIOP-classified marine aerosols. Li et al. (2022) partitioned their CALIPSO-SODA lidar ratios as a function of horizontal averaging and pointed out challenges in CALIOP aerosol typing at longer averages (i.e., 80 km) based on their results. For example, they state "Classification issues for 80 km averaged samples are likely, as spatial averaging are performed to increase the SNR for tenuous aerosol layers, rendering more uncertain retrievals than its 5 and 20 km counterparts." This helped motivate our study to limit the analysis to only those profiles in which at least part of the aerosol layer was detected at 5 km. We have modified the text in the revised manuscript.

Section 4.1: Some details regarding the GEOS GOCART model could have been included earlier in the methods section.

Response: Thank you to the reviewer for this suggestion, however we believe the paper flows better if we include the GOCART model description and methods in the discussion provided in Section 4.1.

Lines 488-489: Please rephrase, is "replacing" indeed the right word?

Response: We have changed "replacing" to "becoming more dominant than".

Line 496: "Eq." is missing before the parenthesis.

Response: Based on the Copernicus style guide, "Eq." should not be placed before the parentheses when defining the equation, but rather when referencing it in the text (as we do in a previous line).

Lines 526 and 535: How was the minimum of 50 points selected?

Response: This threshold was selected after conducting sensitivity studies to ensure a statistically robust characterization of the lidar ratio, while also accounting for satellite data coverage seasonally within each grid box over the study period. We have added this statement to the revised manuscript.

Line 553: The authors could discuss the meteorological conditions leading to the seasonal aerosol transport.

Response: For the sentence in question, we were referring to the Indian monsoon pattern, and the associated prevailing seasonal wind patterns and aerosol transport. These topics are discussed more thoroughly in the Bay of Bengal and Arabian Sea case study of Section 4.4. We have modified the sentence to the following: "These patterns are indicative of seasonal aerosol transport based on the global atmospheric circulation simulated by the GOCART model, including the Indian monsoon (as discussed in more detail in Sect. 4.4)."

Lines 600-606: Could you please elaborate more on the additional procedures, especially on the second one? The minimum lidar ratio of 15 sr is justified from the field measurements. What drove you into implementing the outlier replacement procedure? Where there many outliers and could you please include a statement regarding that?

Response: The outlier replacement procedure was implemented to address some significant discontinuities observed in earlier test versions of the lidar ratio maps. However, there were not many of these cases. For DJF, MAM, and SON, outlier values consisted for ~1% for all grid boxes over water. For JJA, the frequency increased to ~2%. This is discussed further in the paper during the narrative concerning the lidar ratio method flag maps (Fig. 12). We have added the following statement to the revised manuscript:

"This was done to address some significant discontinuities observed in earlier test versions of the $S_a$ maps. However, they only accounted for ~1-2% of all grid boxes over water (Fig. 12)."

Lines 618-619: The reported maximum lidar ratios are clearly influenced by non-marine aerosol and they should be discussed together with the SSVF.

Response: The maximum value for MAM was mistakenly provided as 57 sr but the correct value is 56 sr. It is a model-assisted value in the Bohai Sea (near China), corresponding to a SSVF of 3.5%. The maximum value for JJA is modeled as 57 sr, located in the Caspian Sea (Middle East) and corresponds to a SSVF of 2.5%. The maximum values for SON and DJF are retrievals near the coast in the northern Bay of Bengal and thus are not influenced by modeled SSVF. We have added this discussion to the revised manuscript.

Lines 626-629 and Fig. 11: A lot of pixels flagged as "Retrieval" in Fig. 12 are accompanied by the maximum assigned uncertainty of 22% (e.g., South Atlantic and Pacific during SON). Could you please provide a statement with the typical range of the uncertainty (before the assignment of 22%) for the pixels with the assigned 22% uncertainty?

Response: The reasoning for our use of a maximum assigned uncertainty of 22% is maintaining heritage with the CALIPSO Version 4 products (Kim et al., 2018) and operational expediency due to the ending of the CALIPSO satellite mission. We have added the following statement to the manuscript to address the typical range of uncertainty for the grid cells in question:

"Note that for those grid cells with retrievals and an assigned uncertainty of 22%, the uncertainty median ± uncertainty MAD prior to assignment is 25% ± 2% (DJF and MAM) and 26% ± 2% (JJA and SON)."

Lines 690-695: How and why were the study regions defined as such? It should be stated that by this selection regions with e.g., modelled-only lidar ratios, high SSVF-low lidar ratios etc. were covered.

Response: The study regions were chosen to capture different model scenarios (e.g., coastal versus open oceans) and different derived lidar ratio regimes (e.g., model-assisted vs. retrieval).  The following sentence was added to the paper to reflect this:

"These regions were selected specifically to capture different aerosol model scenarios, including coastal (typically low SSVF, thus higher $S_a$) and open oceans (typically high SSVF, thus lower $S_a$), and various derived-$S_a$ regimes in general (e.g., model versus retrieval)."

Lines 746-747: For clarity, please point out again that these results correspond to 2015.

Response: We have made this change as suggested.

References:

Ansmann, A., Wagner, F., Althausen, D., Müller, D., Herber, A., & Wandinger, U. (2001). European pollution outbreaks during ACE 2: Lofted aerosol plumes observed with Raman lidar at the Portuguese coast. *Journal of Geophysical Research: Atmospheres*, *106*(D18), 20725-20733, https://doi.org/10.1029/2000JD000091.

Burton, S. P., Ferrare, R. A., Vaughan, M. A., Omar, A. H., Rogers, R. R., Hostetler, C. A., and Hair, J. W.: Aerosol classification from airborne HSRL and comparisons with the CALIPSO vertical feature mask, Atmos. Meas. Tech., 6, 1397–1412, https://doi.org/10.5194/amt-6-1397-2013, 2013.

Dawson, K. W., Meskhidze, N., Josset, D., & Gassó, S. (2015). Spaceborne observations of the lidar ratio of marine aerosols. *Atmospheric Chemistry and Physics*, *15*(6), 3241-3255, https://doi.org/10.5194/acp-15-3241-2015.

Groß, S., Gasteiger, J., Freudenthaler, V., Müller, T., Sauer, D., Toledano, C., & Ansmann, A. (2016). Saharan dust contribution to the Caribbean summertime boundary layer–a lidar study during SALTRACE. *Atmospheric Chemistry and Physics*, *16*(18), 11535-11546, https://doi.org/10.5194/acp-16-11535-2016.

Kim, M. H., Omar, A. H., Tackett, J. L., Vaughan, M. A., Winker, D. M., Trepte, C. R., ... & Magill, B. E. (2018). The CALIPSO version 4 automated aerosol classification and lidar ratio selection algorithm. *Atmospheric measurement techniques*, *11*(11), 6107-6135, https://doi.org/10.5194/amt-11-6107-2018.

Kuciauskas, A. P., Xian, P., Hyer, E. J., Oyola, M. I., & Campbell, J. R. (2018). Supporting weather forecasters in predicting and monitoring Saharan air layer dust events as they impact the greater Caribbean. *Bulletin of the American Meteorological Society*, *99*(2), 259-268, https://doi.org/10.1175/BAMS-D-16-0212.1.

Li, Z., Painemal, D., Schuster, G., Clayton, M., Ferrare, R., Vaughan, M., ... & Trepte, C. (2022). Assessment of tropospheric CALIPSO Version 4.2 aerosol types over the ocean using independent CALIPSO–SODA lidar ratios. *Atmospheric Measurement Techniques*, *15*(9), 2745-2766, https://doi.org/10.5194/amt-15-2745-2022.

Liu, Z., Omar, A., Vaughan, M., Hair, J., Kittaka, C., Hu, Y., ... & Pierce, R. (2008). CALIPSO lidar observations of the optical properties of Saharan dust: A case study of long-range transport. *Journal of Geophysical Research: Atmospheres*, *113*(D7), https://doi.org/10.1029/2007JD008878.

Oo, M., & Holz, R. (2011). Improving the CALIOP aerosol optical depth using combined MODIS-CALIOP observations and CALIOP integrated attenuated total color ratio. *Journal of Geophysical Research: Atmospheres*, *116*(D14), https://doi.org/10.1029/2010JD014894.

Rogers, R. R., Vaughan, M. A., Hostetler, C. A., Burton, S. P., Ferrare, R. A., Young, S. A., ... & Winker, D. M. (2014). Looking through the haze: evaluating the CALIPSO level 2 aerosol optical depth using airborne high spectral resolution lidar data. *Atmospheric Measurement Techniques*, *7*(12), 4317-4340, https://doi.org/10.5194/amt-7-4317-2014.

Welton, E. J., Voss, K. J., Quinn, P. K., Flatau, P. J., Markowicz, K., Campbell, J. R., ... & Johnson, J. E. (2002). Measurements of aerosol vertical profiles and optical properties during INDOEX 1999 using micropulse lidars. *Journal of Geophysical Research: Atmospheres*, *107*(D19), INX2-18, https://doi.org/10.1029/2000JD000038.

---

## Author Comment (AC2)

"Mapping 532 nm Lidar Ratios for CALIPSO-Classified Marine Aerosols using MODIS AOD Constrained Retrievals and GOCART Model Simulations" by Toth et al. documents the updates to the lidar ratio selection methodology for marine aerosols in v5. The updated method uses MODIS AODs and GOCART modeled aerosols to create seasonally and spatially varying maps from which to select their marine aerosol lidar ratios. They find that these updates provide AODs that better align with those calculated through the ODCOD than the previous version (v4.51) and also better agree with those measured by AERONET sites in coastal and island locations. The paper provides valuable documentation of the updated CALIOP data product, which, despite CALIPSO's retirement in 2023, still provides a valuable long-term dataset for cloud/aerosol research. This update to marine aerosol lidar ratios represents a significant advancement through addressing regional and seasonal variability that was previously unaccounted for in the previous fixed value lidar ratio assignment. The paper is generally well organized and written; however, this reviewer found it to be a bit on the long side. I would recommend publication after some minor revisions.

Response: We thank the reviewer for the helpful feedback and comments, which have contributed to strengthening this manuscript.

Major Points:

1. The study provides a valuable update to the assignment of marine aerosol lidar ratios; however, the approach raises fundamental questions about CALIOP's aerosol typing framework. The manuscript would benefit from directly addressing how these new spatially/seasonally varying marine lidar ratios relate to the existing aerosol typing framework, particularly the distinction between marine and dusty marine. Is differentiating between dusty marine and marine needed or useful anymore with these new methods?

Response: Good question!  Our answer is clearly "yes", as identifying (and also quantifying) the dust content in any aerosol plume remains a topic of scientific interest.  For an offhand bit of evidence, we point to the increased usage of the LIVAS product developed by the National Observatory of Athens (e.g., see https://acp.copernicus.org/articles/22/535/2022/).

Many of the regions where the largest differences in lidar ratios occur, such as the Bay of Bengal, are regions where dusty marine and other aerosol mixtures are common. Here the study assigned V5 marine lidar ratios exceeding the V4.51 dusty marine value of 37 sr in some regions. This convergence between the new variable marine lidar ratios and the dusty marine values raises two questions for me: 1) may some of these aerosol layers currently classified as marine in these regions be misclassified dusty marine?

Response: Yes, of course.  The CALIPSO aerosol classification scheme is not perfect.  But so long as the optical depths above the layers being classified are fairly low (or, as in the case of this study, zero), the separation between marine and dusty marine is robust, as it relies primarily on the estimated particulate depolarization ratio of the layer (Kim et al., 2018).

There may, however, be some misclassification of marine as dusty marine in cases of very low humidities where marine aerosols can transition from droplets to desiccated sea salt crystals (Ferrare et al., 2023).

2) Does this new method render the discernment between marine and dusty marine somewhat obsolete?

Response: No, not at all; e.g., see Groß et al., 2013, who show the changes in lidar ratio and particulate depolarization ratio as a function of dust fraction within a layer. Changes in dust fraction are reflected in changes in depolarization, which can then be mapped into changes in lidar ratio for dusty marine mixes.

Connecting lidar ratios to modeled sea salt volume fractions suggests that this approach could be beneficially extended to other marine-influenced aerosol classes, which perhaps is covered by the tables/maps noted at L142, but not shown in this paper.

Response: The seasonal maps of marine lidar ratios presented in this paper (and included in V5) do not impact the overall aerosol typing framework, including the distinction between marine and dusty marine. There have been no changes from V4 to the V5 aerosol typing classification algorithms as it pertains to distinguishing between these two CALIOP aerosol types (details of the classification algorithm are included in Fig. 1 in the manuscript and discussed in detail in Kim et al. 2018).

Regarding the question on whether or not differentiating between these two types is needed or useful given the new V5 methods, we believe that this is indeed necessary. In the early stages of our analysis, we did not find a clear difference in the MODIS AOD constrained lidar ratio retrievals between marine-only and dusty marine-only CALIOP profiles. When focusing on "the Atlantic dust corridor" for the period between 2006 through 2017, dusty marine lidar ratios were higher than marine lidar ratios by ~1.8 sr. However, in a later study conducted over the same region but with more rigorous data selection criteria, we found dusty marine lidar ratios exceeded marine lidar ratios by just over 3 sr. Adapting the method given in Groß et al., 2013 shows that this 3 sr difference is equivalent to a dust fraction of ~25%. Because CALIOP depolarization measurements are robust and layer integrated depolarization is a strong indicator of dust fraction, we decided to create separate lidar ratio maps for marine and dusty marine, rather than combine them. Due to our desire to optimize the content and flow of this manuscript, we purposely only focused on the marine maps here, and expect to focus on dusty marine in a separate paper.

Responding to the reviewer's suggestion that this approach could be beneficially extended to other marine-influenced aerosol classes:

Unfortunately, it's difficult to see how we could easily extend this approach to other "marine-influenced aerosol classes". Empirically deriving lidar ratios according to aerosol type requires a highly accurate aerosol type discrimination scheme. Ideally, one would construct such a scheme by measuring several intrinsic properties of the aerosol such as depolarization ratios, color ratios, and lidar ratios, then use these quantities to infer aerosol type. (Note that we've put the cart before the horse here; the intrinsic properties approach assumes that lidar ratios can be either directly measured or trivially retrieved from the direct measurements.) While multi-wavelength HSRLs excel at this (e.g., Burton et al., 2014), elastic backscatter lidars like CALIOP simply cannot perform the same magic. Instead of determining aerosol type base on measured intrinsic properties, CALIOP must use extrinsic properties to determine aerosol type (Vaughan et al., 2021). Then, having determined type, look up tables are used to *assign* the

lidar ratios that are subsequently used to *calculate* estimates of the same intrinsic properties that an HSRL can measure. For more discussion on this point, see Burton et al., 2014.

Depolarization ratios provide the single case in which extrinsic properties can (usually) be used to accurately discriminate aerosol types. CALIOP's *estimated* particulate depolarization ratio combines measurements of layer integrated volume depolarization with estimates of the particulate optical depth overlying any layer to approximate the *true* particulate depolarization ratio. As noted earlier, when the overlying optical depths are low, this approximation is gratifyingly accurate and hence a highly reliable metric for identifying aerosol layers with non-zero dust fractions. The discriminatory power of the *estimated* particulate depolarization ratios is illustrated below in Figure 1, which was produced during one of the many sensitivity studies conducted for this paper.

[Figure]

Figure 1: distribution of retrieved layer-integrated particulate depolarization ratios for all measurements in the Atlantic dust corridor during 2010 through 2017. The data in this study was restricted to those profiles averaged to 5-km along track resolution in which only a single layer was detected. Both nighttime and daytime measurements are included. Note that the CALIOP aerosol subtyping algorithm defines a depolarization threshold of 0.075 to separate marine aerosol from dusty marine, and this explains the sharp partitioning of the distributions that occurs at that value.

[Figure]

Figure 2: coordinates of the Atlantic dust corridor used to harvest the data in Figure 1.

The only other measured quantity that CALIOP might conceivably use as a proxy for an intrinsic property is the layer-integrated total attenuated backscatter color ratio, $\chi'$. As a substitute for an Ångström exponent, this quantity might be expected to yield information about aerosol size composition and hence give insights into aerosol type. However, as illustrated in Figure 3 and noted in Kim et al., 2018, $\chi'$ shows no skill in differentiating between the seven CALIOP tropospheric aerosol type, as the $\chi'$ frequency distributions for all types lie more-or-less on top of one another.

[Figure]

Figure 3: Distributions of layer-integrated total attenuated backscatter color ratio partitioned by aerosol subtype for nighttime measurements of the uppermost layer in a 5 km column acquired during 2013–2015 (shamelessly pillaged from Vaughan et al., 2021).

2. The exclusion of modeled dust aerosols from the SSVF calculations (L407-409) warrants reconsideration or further justification. This study already focuses specifically on CALIOP-identified marine layers, which have already passed the CALIOP typing algorithm's criteria (depolarization or otherwise) for classification as marine vs. dust (or other types). These marine-classified layers would still contain some dust at concentrations below levels that would trigger classification as dusty marine or something else. By

omitting dust from the SSVF denominator, SSVFs would be inflated, especially in transitional, dustier, regions.

Response: As we state in the paper, the primary reason we chose to exclude dust from our SSVF calculations was due to the ability of CALIOP to classify dust through the depolarization ratio. However, we recognize that some residual dust may still be included in the CALIOP-classified marine aerosol layers and agree that the inclusion of dust in our SSVF computations warrants consideration. As part of the extensive analyses required in these modeling-related efforts, we briefly assessed how the SSVFs and corresponding model-assisted lidar ratios would change by including dust for the entire study period (rather than partitioning by year and/or season). While it is true that in dust-prone areas there are reduced SSVFs when including dust, the second order polynomial equation (Equation 1) characterizing the relationship between SSVF and retrieved lidar ratio also changes. This results in a near-zero change in model-assisted lidar ratios for most of the global oceans. The exception is the dust belt in the Atlantic Ocean, for which there could be lidar ratios ~13 sr larger by including dust (however, these cases would be classified as dusty marine or dust; see Fig. 1). We note, though, that any change in SSVF will only impact the model-assisted lidar ratios, not the retrievals (as shown in Fig. 12). When considering this, the area with the most impact by including dust would be the modeled values in the Atlantic dust belt in the JJA season. While it is ideal to further investigate this to fully understand the potential changes, this would require more extensive evaluation, and the implementation of updated lidar ratio maps would require reprocessing of the CALIPSO V5 data products. Due to the closeout of the CALIPSO satellite mission in September 2025, and the simultaneous loss of funding for all members of the CALIPSO project team, this will unfortunately not be possible.

3. The manuscript is a bit lengthy and could be strengthened through some editing, especially in the introduction. The extensive literature review from L148 to 222 largely duplicates the information already presented and effectively summarized in Table 2. Streamlining the literature review and highlighting only the most significant studies would benefit readers.

Response: We agree with the reviewer that the manuscript is lengthy, particularly in the Introduction during the literature review. As found in the revised manuscript, we have shortened this section by removing several lines of discussion and highlighted just a few studies. The results from other papers are summarized in Table 2.

Minor Points:

L164: is the MPL at 532 or 523nm?

Response: We double-checked the corresponding papers and can confirm that the MPLs for these studies operated at a wavelength of 523 nm, not 532 nm.

L330: This sentence read a bit weird. Consider: "Note that this approach produces a negligible proportion of negative Sa values (less than 0.05%), and our methodology minimizes the influence

of these outliers by using median values when creating the Sa maps (Sections 3 and 4)." or something similar. The phrase "our use of medians" sounded off to me…

Response: We have edited this sentence as suggested.

L347: What are typical stratospheric AOD values in the SAPP? It strikes me that removing stratospheric AOD from the column would result in a fairly small correction outside of volcanic/pyrocumulonimbus events.

Response: The stratospheric AODs reported in the SAPP are typically < 0.01, and these were shown to agree generally well with Stratospheric Aerosol and Gas Experiment III (SAGE III) measurements between about 30° S and 30° N (Kar et al., 2019; Li et al., 2022). For our constrained lidar ratio analysis, the global mean stratospheric AOD used was ~0.009, with a global median value of ~0.007. Our sensitivity studies as part of this work resulted in a ~2 sr reduction in lidar ratio globally when accounting for these stratospheric AOD values.

Figure 13: The regional boxes encompass a lot of land. I would recommend being more explicit that the analysis only includes at the oceanic parts of the domain.

Response: Thank you for this suggestion. We have added the following sentence after the first mention of Fig. 13: "While some regions encompass a large amount of land, only the oceanic parts of each domain are used in the analysis."

L763: State why ODCOD is expected to be greater than v4.51

Response: In the previous paragraph to the sentence in question, we discuss the reasons for the differences between the standard CALIOP retrieval and ODCOD (i.e., $S_a$ selection and layer detection). We have added the following sentence to that paragraph, referring specifically to layer detection: "This can be due to optically thin layers that are below CALIOP's direct detection thresholds and are not detected as features in the standard retrieval but are responsible for attenuation that is accounted for in the ODCOD retrieval." We have also added "i.e., due at least partly to layer detection" to explain why ODCOD is expected to be greater than V4.51.

L816: Consider adding that models parameterize sea salt emissions by wind speed.

Response: We have added this statement to the paper.

References:

Burton, S. P., R. A. Ferrare, M. A. Vaughan, A. H. Omar, R. R. Rogers, C. A. Hostetler, and J. W. Hair, 2013: Aerosol Classification from Airborne HSRL and Comparisons with the CALIPSO Vertical Feature Mask, *Atmos. Meas. Tech.*, **6**, 1397–1412, https://doi.org/10.5194/amt-6-1397-2013.

Burton, S. P., M. A. Vaughan, R. A. Ferrare, and C. A. Hostetler, 2014: Separating mixtures of aerosol types in airborne High Spectral Resolution Lidar data, *Atmos. Meas. Tech.*, **7**, 419–436, https://doi.org/10.5194/amt-7-419-2014.

Ferrare, R. and 27 coauthors, 2023: Airborne HSRL-2 measurements of elevated aerosol depolarization associated with non-spherical sea salt, *Front. Remote Sens.*, **4**, 1143944, https://doi.org/10.3389/frsen.2023.1143944.

Groß, M. S., Esselborn, B. Weinzierl, M. Wirth, A. Fix and A. Petzold, 2013: Aerosol classification by airborne high spectral resolution lidar observations, *Atmos. Chem. Phys.*, **13**, 2487–2505, https://doi.org/10.5194/acp-13-2487-2013.

Kar, J., Lee, K. P., Vaughan, M. A., Tackett, J. L., Trepte, C. R., Winker, D. M., ... & Getzewich, B. J. (2019). CALIPSO level 3 stratospheric aerosol profile product: version 1.00 algorithm description and initial assessment. *Atmospheric Measurement Techniques*, *12*(11), 6173-6191, https://doi.org/10.5194/amt-12-6173-2019.

Kim, M.-H., A. H. Omar, J. L. Tackett, M. A. Vaughan, D. M. Winker, C. R. Trepte, Y. Hu, Z. Liu, L. R. Poole, M. C. Pitts, J. Kar, and B. E. Magill, 2018: The CALIPSO Version 4 Automated Aerosol Classification and Lidar Ratio Selection Algorithm, *Atmos. Meas. Tech.*, **11**, 6107-6135, https://doi.org/10.5194/amt-11-6107-2018.

Li, Z., Painemal, D., Schuster, G., Clayton, M., Ferrare, R., Vaughan, M., ... & Trepte, C. (2022). Assessment of tropospheric CALIPSO Version 4.2 aerosol types over the ocean using independent CALIPSO–SODA lidar ratios. *Atmospheric Measurement Techniques*, *15*(9), 2745-2766, https://doi.org/10.5194/amt-15-2745-2022.

Vaughan, M., M. Kacenelenbogen, J. Tackett, J. Kar, R. Ryan, A. Omar, S. Young, X. Lu, J. Reagan and G. Schuster, 2021: Exploiting 1064 nm Measurements to Improve CALIOP Aerosol Type Identification, presentation given at the CALIPSO Version 5 Aerosol Lidar Ratio Workshop held virtually March 9-11, 2021; slides available at https://science.larc.nasa.gov/wp-content/uploads/sites/147/2021/04/2021-CALIPSO-V5-ALR-Workshop-Vaughan_Mark.pptx